# Tight nuclear tethering of cGAS is essential for preventing autoreactivity

**Hannah E Volkman[†], Stephanie Cambier[†], Elizabeth E Gray[‡], Daniel B Stetson***

Department of Immunology, University of Washington School of Medicine, Seattle, United States

**Abstract** cGAS is an intracellular innate immune sensor that detects double-stranded DNA. The presence of billions of base pairs of genomic DNA in all nucleated cells raises the question of how cGAS is not constitutively activated. A widely accepted explanation for this is the sequestration of cGAS in the cytosol, which is thought to prevent cGAS from accessing nuclear DNA. Here, we demonstrate that endogenous cGAS is predominantly a nuclear protein, regardless of cell cycle phase or cGAS activation status. We show that nuclear cGAS is tethered tightly by a salt-resistant interaction. This tight tethering is independent of the domains required for cGAS activation, and it requires intact nuclear chromatin. We identify the evolutionarily conserved tethering surface on cGAS and we show that mutation of single amino acids within this surface renders cGAS massively and constitutively active against self-DNA. Thus, tight nuclear tethering maintains the resting state of cGAS and prevents autoreactivity.

**\*For correspondence:**
stetson@uw.edu

[†]These authors contributed equally to this work

**Present address:** [‡]Seattle Genetics, Bothell, United States

**Competing interests:** The authors declare that no competing interests exist.

## Introduction

The cGAS-STING DNA sensing pathway has emerged as a key innate immune response that is important for antiviral immunity (*Goubau et al., 2013*), contributes to specific autoimmune diseases (*Crowl et al., 2017*), and mediates important aspects of antitumor immunity (*Li and Chen, 2018*). cGAS binds to double-stranded DNA and catalyzes the formation of cyclic GMP-AMP (*Sun et al., 2013*; *Wu et al., 2013*), a diffusible cyclic dinucleotide that activates the endoplasmic adaptor protein STING (*Ishikawa et al., 2009*). Activated STING then serves as a platform for the inducible recruitment of the TBK1 kinase, which phosphorylates and activates the transcription factor IRF3, leading to the induction of the type I interferon mediated antiviral response (*Liu et al., 2015*).

cGAS is important for 'cytosolic DNA sensing,' a term that was first proposed in 2006, years before the discovery of STING and cGAS as the essential adaptor and unique sensor of this pathway (*Stetson and Medzhitov, 2006*). At the time, a key conundrum was how a sequence-independent DNA sensing pathway that was activated by the sugar-phosphate backbone of DNA could avoid constant autoreactivity against genomic DNA that is present in all nucleated cells. We proposed the possibility that the sensor would be sequestered in the cytosol, separated by the nuclear envelope from genomic DNA, and that the inappropriate appearance of DNA in the cytosol would enable detection of foreign DNA while maintaining 'ignorance' to self DNA (*Stetson and Medzhitov, 2006*). The discovery of cGAS and cGAMP, and subsequent structural studies of cGAS binding to DNA, provided an elegant explanation for the sequence independence of the response, the requirement for double-stranded DNA as a ligand, the contribution of the deoxyribose sugar-phosphate backbone of DNA to detection, and the definitive link between DNA sensing and STING (*Civril et al., 2013*; *Li et al., 2013*; *Sun et al., 2013*; *Wu et al., 2013*). However, the precise location of cGAS prior to its activation has remained largely unexplored, in part because of a lack of tools to track endogenous cGAS. Cytosolic DNA sensing has persisted as the mechanistic framework that guides the field.

There are a number of important problems with the model of cytosolic DNA sensing. First, nearly all DNA viruses (with the exception of the poxviruses) replicate their DNA exclusively in the nucleus. Studies of cGAS-deficient mouse and human cells have revealed that cGAS is important for the IFN-mediated antiviral response to these nuclear-replicating viruses, including herpesviruses (*Ma et al., 2015*; *Wu et al., 2015*). Moreover, retroviruses and lentiviruses are detected by cGAS (*Gao et al., 2013*; *Lahaye et al., 2013*; *Rasaiyaah et al., 2013*), but they shield their DNA within a capsid during reverse transcription in the cytosol, releasing this DNA for integration into the genome upon translocation into the nucleus. To fit the concept of cytosolic DNA sensing, current models envision that such viruses 'leak' DNA into the cytosol during cellular entry or during exit. Second, as noted in the original description of cytosolic DNA sensing (*Stetson and Medzhitov, 2006*), cell division results in the breakdown of the nuclear envelope and the mixing of cytosolic and nuclear contents, which challenges a simple model of cytosolic sequestration as the basis for self/non-self discrimination by cGAS. Indeed, recent studies have demonstrated that cGAS can be found associated with mitotic chromosomes (*Yang et al., 2017*). This association is thought to be mediated by the generic DNA binding properties of cGAS, and it has been proposed that upon resolution of cell division and reformation of the nuclear envelope, cGAS is redistributed to the cytosol (*Yang et al., 2017*).

Here, we use confocal microscopy and biochemical characterization to determine the resting localization of endogenous cGAS prior to activation. We unexpectedly find that the vast majority of cGAS is in the nucleus, regardless of whether cells are rapidly dividing or post-mitotic. Moreover, we demonstrate that cGAS is tethered tightly in the nucleus by a salt-resistant interaction that rivals that of histones in its strength. Finally, we identify the tethering surface on cGAS and show that tight nuclear tethering prevents cGAS autoreactivity against self-DNA.

## Results

### Endogenous cGAS is predominantly a nuclear protein

We screened numerous commercially available antibodies to human cGAS for their ability to identify endogenous cGAS unambiguously and specifically using immunofluorescence microscopy. We chose to image HeLa cells, which express endogenous cGAS that is inactive in resting cells and potently activated to produce cGAMP upon transfection of calf thymus DNA (*Figure 1—figure supplement 1A*). Despite this potent activation of cGAS and production of cGAMP after DNA transfection, HeLa cells fail to activate the type I interferon response because the E7 oncoprotein of human papillomavirus 18 blocks STING-dependent signaling (*Lau et al., 2015*). To test for specificity of staining, we used lentiCRISPR to generate clonal lines of cGAS-deficient HeLa cells (*Figure 1—figure supplement 1B*; (*Gray et al., 2016*). We found that the D1D3G rabbit monoclonal antibody that detects an epitope in the N terminus of human cGAS was suitable for analysis of endogenous cGAS by microscopy. Unexpectedly, endogenous cGAS was localized almost exclusively in the nuclei of all HeLa cells, with little cytosolic staining (*Figure 1A*, *Figure 1—videos 1* and *2*). Identically prepared cGAS-deficient HeLa cells had no detectable immunostaining, confirming the specificity of this antibody for endogenous cGAS (*Figure 1A*). We noted three additional reproducible patterns of cGAS localization in addition to the uniform nuclear staining. First, as observed previously (*Yang et al., 2017*), we found that cGAS was associated with condensed mitotic chromatin (*Figure 1B*). Second, we found cGAS in rare, spontaneous, DAPI-positive, micronucleus-like extranuclear structures (*Figure 1B*). Whereas cGAS localization to micronuclei has been reported recently in a number of studies that primarily visualized overexpressed cGAS (*Bartsch et al., 2017*; *Dou et al., 2017*; *Glück et al., 2017*; *Harding et al., 2017*; *Mackenzie et al., 2017*; *Yang et al., 2017*), we found that all cells with such structures also had extensive endogenous cGAS staining in the main nucleus (*Figure 1B*). Third, we found endogenous cGAS localized to 'chromatin bridges' between adjacent cells (*Figure 1B*), the origins of which are thought to involve chromosome fusions and incomplete segregation of DNA between daughter cells during mitosis (*Maciejowski et al., 2015*).

To extend our findings to primary cells of another species, we searched for antibodies that could identify endogenous mouse cGAS by microscopy. Using primary bone marrow-derived macrophages (BMMs) from wild-type and cGAS-deficient mice and a mouse-specific cGAS antibody, we found nearly exclusive localization of mouse cGAS to the nucleus (*Figure 1C*). However, even with optimization of blocking conditions and antibody dilutions, we noted that cGAS-deficient mouse

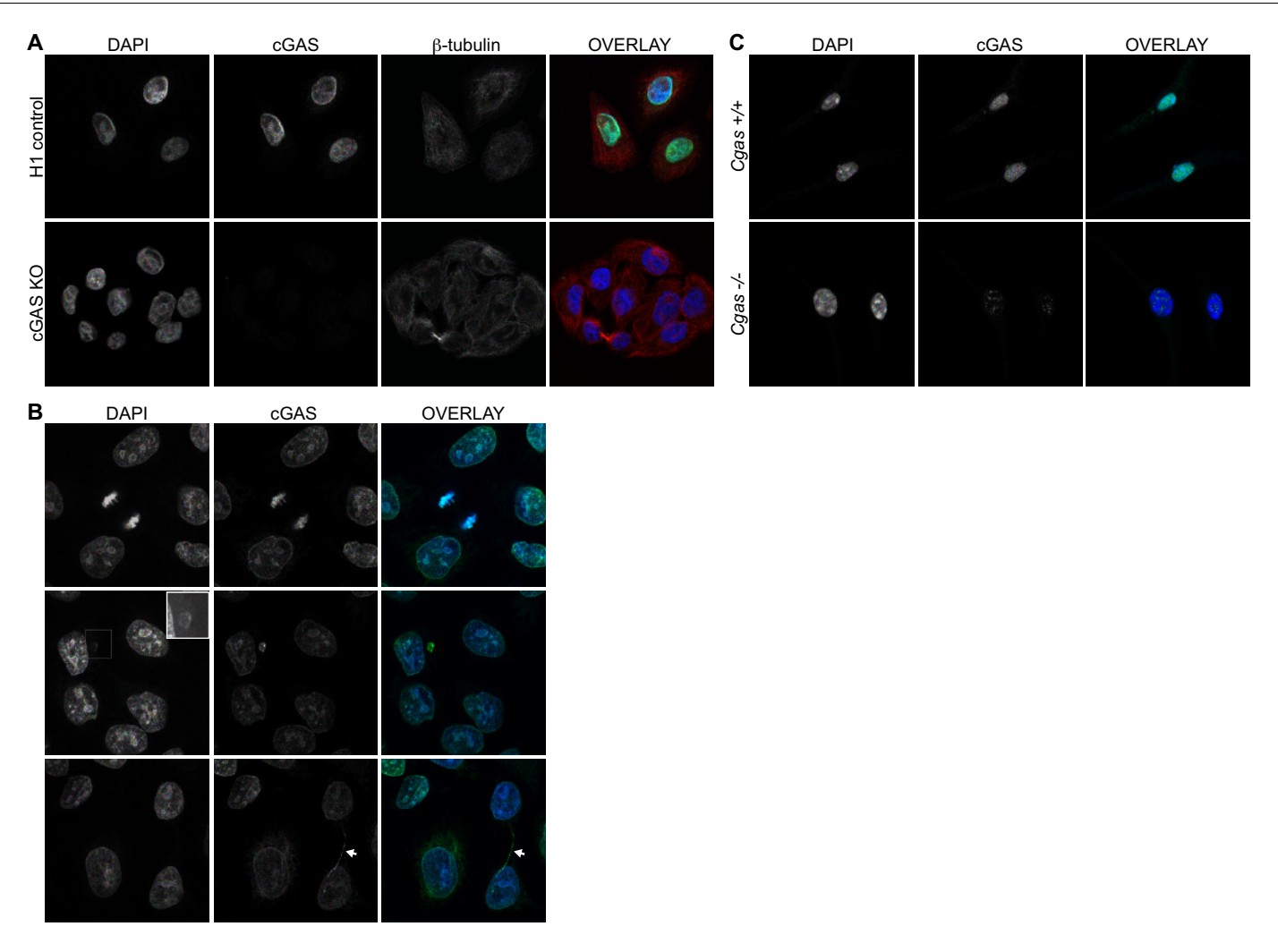

**Figure 1.** Endogenous cGAS is predominantly a nuclear protein. (**A**) Clonal lines of HeLa cells were generated using lentiCRISPR encoding either a non-targeting H1 control gRNA (top row) or a cGAS-targeted gRNA (cGAS KO). Cells were fixed with methanol, stained with antibodies to human cGAS and beta-tubulin, counter-stained with DAPI, and visualized by confocal microscopy. (**B**) We noted three reproducible patterns of cGAS localization in addition to the nucleus: condensed mitotic chromatin (top row), structures resembling micronuclei (middle row), and tendril-like bridges between cells. (**C**) Mouse *Cgas+/+* and *Cgas-/-* primary bone marrow-derived macrophages were stained using a mouse antibody to cGAS and processed as in (**A**).

The online version of this article includes the following video and figure supplement(s) for figure 1:

**Figure supplement 1.** Characterization of clonal cGAS KO HeLa cells and microscopy conditions for cGAS visualization.

**Figure 1—video 1.** Compiled z-series confocal images of the HeLa cells visualized in *Figure 1A*.

https://elifesciences.org/articles/47491#fig1video1

**Figure 1—video 2.** Rendered z-series of HeLa cells rotated around the y-axis.

https://elifesciences.org/articles/47491#fig1video2

macrophages displayed a pattern of nuclear staining that was distinct in its distribution and less abundant than the cGAS staining of wild-type cells (*Figure 1C*). Despite the imperfect background fluorescence, this was the most sensitive and specific cGAS staining we could identify among the antibodies that we tested.

In our microscopy experiments, we used methanol fixation/extraction because we found that the epitope of the antibody to human cGAS was sensitive to paraformaldehyde (PFA) fixation, which reduced the signal of the specific staining and increased background staining. However, it has been suggested that methanol fixation might also extract a membrane-bound pool of cytosolic cGAS

(*Barnett et al., 2019*), leading to an overestimation of the amount of nuclear cGAS in our images. We found that PFA fixation followed by 0.1% Triton X-100 permeabilization, which would preserve such a pool of cGAS, resulted in a pattern of endogenous cGAS staining that was almost exclusively nuclear and nearly identical to the staining observed in methanol-fixed cells (*Figure 1—figure supplement 1C*). Finally, to rule out any role for fixation in our interpretation of cGAS localization, we performed Amnis imaging flow cytometry on live, unfixed cGAS KO HeLa cells stably expressing a GFP-cGAS fusion protein. Analysis of thousands of individual cells revealed that the great majority of cGAS colocalized with a fluorescent DNA-intercalating dye that marked the nucleus (*Figure 1—figure supplement 1D*). Together, these data reveal that, contrary to expectation, cGAS is primarily a nuclear protein in both human and mouse cells.

## cGAS is tethered tightly in the nucleus

We sought to reconcile the nuclear localization of endogenous cGAS in *Figure 1* with the widely accepted notion that cGAS is primarily a cytosolic protein. To track endogenous cGAS localization thoroughly, we modified a protocol for salt-based elution of histones from purified nuclei (*Shechter et al., 2007*). We prepared extracts separating cytosol from nuclei using a solution containing 0.2% NP-40 detergent followed by low speed centrifugation. After washing the pellets with detergent-free lysis buffer, we lysed the nuclei in a solution of 3 mM EDTA and 0.3 mM EGTA in water. Following this zero salt nuclear lysis, the pellets remaining after centrifugation were treated with stepwise increases of NaCl in a buffer containing 50 mM Tris-HCl pH 8.0 and 0.05% NP-40. We tracked endogenous cGAS throughout this sequential extraction and elution protocol using six different cell lines from humans and mice, sampling primary cells, immortalized cells, and tumor cells. These included cells that were actively dividing (HeLa, SiHa, mouse fibroblasts, human fibroblasts) as well as primary mouse macrophages that are largely non-dividing (*Luo et al., 2005*). We monitored the specificity of the extractions using the cytosolic protein tubulin, the nuclear zero/low salt elution-enriched protein LSD1, and nuclear pellet-localized histones H2B and H3. In all of these cells, we found that the vast majority of cGAS was not only in the nuclear fractions, but it was remarkably resistant to salt-based elution (*Figure 2A*). In most of these cells, a NaCl concentration of 0.75 M or higher was required to solubilize the majority of cGAS, similar to the amount of salt required to initiate the liberation of histones (*Figure 2A*). Importantly, the salt elutions reflect sequential treatments of the same nuclear extracts such that the sum of all the cGAS signals in these fractions can be compared to the cytosolic extracts in order to determine the relative amounts of cGAS in the cytosol and nucleus. These comparisons, calculated by densitometry analysis (*Figure 2B*), corroborate the microscopy studies in *Figure 1* and reveal that the great majority of endogenous cGAS is in the nucleus, not in the cytosol. Moreover, our findings demonstrate that the conventional nuclear washes of ~420 mM NaCl that are typically used to isolate nuclear proteins are insufficient to liberate cGAS from the nucleus (*Sun et al., 2013*). Such tight tethering of cGAS in the nucleus cannot be explained by its low intrinsic affinity for DNA, the dissociation constant of which has been estimated at 1–2 μM (*Civril et al., 2013*; *Li et al., 2013*).

## cGAS is nuclear regardless of cell cycle phase or activation state

One potential explanation for the nuclear localization of cGAS, particularly in dividing cells like HeLa cells, is that this reflects the previously observed association of cGAS with condensed mitotic chromatin. Thus, recently divided cells might still retain cGAS in the nucleus before its redistribution to the cytosol via mechanisms that remain unexplained. Importantly, the fact that almost 95% of cGAS is nuclear in largely post-mitotic, differentiated primary mouse macrophages argues against this possibility (*Figures 1C* and *2*). We further tested this by tracking the localization of endogenous cGAS throughout controlled cell cycles in HeLa cells. Based on our observation that nuclear cGAS is resistant to salt elution up to 0.75 M NaCl (*Figure 2*), we used a widely available commercial extraction kit to separate cytosol from the nuclear proteins that elute in ~420 mM NaCl (nuclear supernatant, NS), and we additionally examined the residual pellets to visualize the entire pool of tightly tethered cGAS (nuclear pellet, NP). We used double-thymidine block to synchronize HeLa cells at the G1/S boundary (*Bootsma et al., 1964*), followed by release that resulted in a uniform progression through a single cell cycle. At 4 and 8 hr post release, PI staining confirmed uniform populations of cells in S and G2/M phases, respectively (*Figure 3A*). By 24 hr, the cells had become asynchronous again

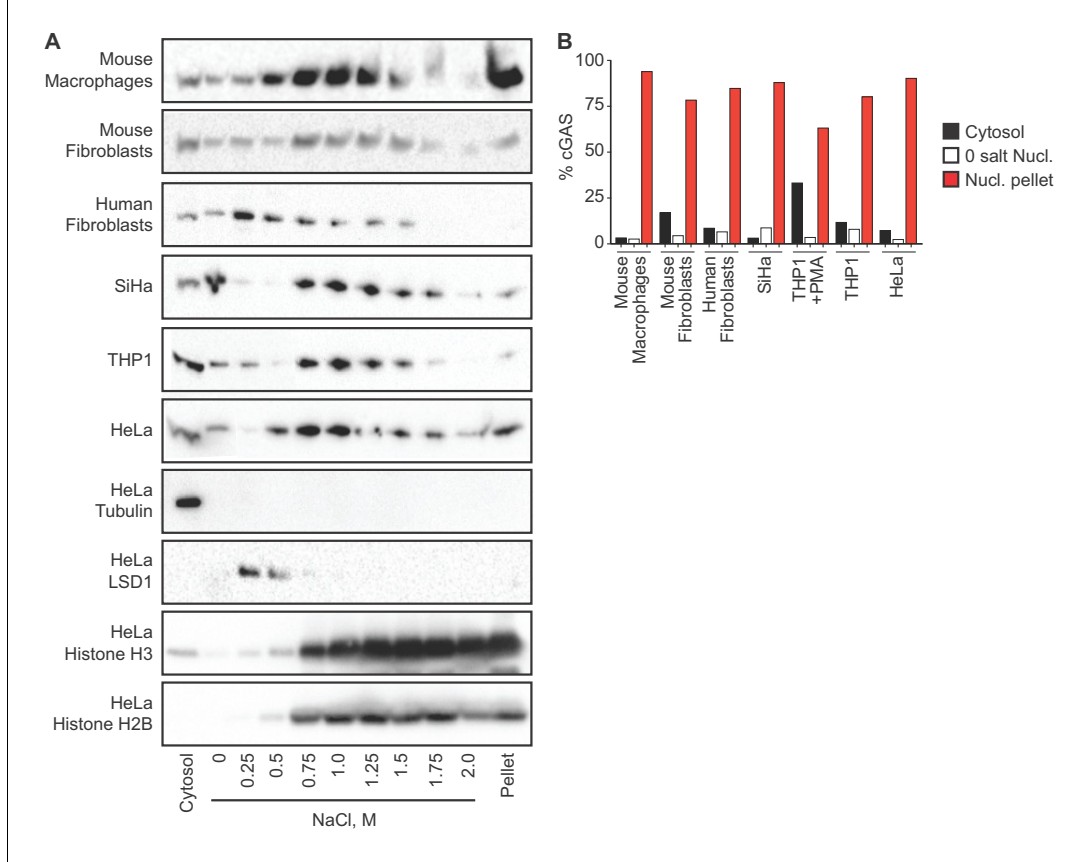

**Figure 2.** cGAS is tightly tethered in the nucleus. (**A**) Mouse and human cell lines were separated into cytosolic and nuclear fractions, followed by sequential stepwise elutions of nuclear pellets with the indicated concentrations of NaCl. cGAS and the indicated control proteins (shown for HeLa cells) were monitored throughout the elution by western blot. (**B**) Densitometry measurements quantitating the relative amounts of endogenous cGAS protein in the cytosol, the 0 salt nuclear lysis, and the cumulative nuclear pellet.

(*Figure 3A*). At all of these time points, we found that the localization of the majority of cGAS to the nuclear pellet did not change (*Figure 3B*). Thus, endogenous cGAS is a tightly tethered nuclear protein, regardless of cell cycle phase.

Next, we asked whether cGAS localization is dependent on its activation state. We transfected HeLa cells with calf thymus DNA and harvested them four hours later, at a time when they were making large amounts of cGAMP (*Figure 1—figure supplement 1A*). We performed sequential extractions and salt elutions, comparing cGAS distribution in control and stimulated cells. We did not observe any concerted relocalization of cGAS into the cytosol, despite its robust activation at this time point (*Figure 3C*). These data demonstrate that activation of cGAS by foreign DNA does not result in a dramatic redistribution to the cytosol.

## cGAS nuclear tethering and cGAS activation are governed by separate mechanisms

We next determined the domains of cGAS that contribute to its tight tethering in the nucleus. The core of human cGAS is comprised of a bilobed nucleotidyltransferase (NTase) structure bridged by an alpha-helical spine (*Figure 4A*) (*Civril et al., 2013*; *Li et al., 2013*). In addition, the N-terminal ~150 residues of cGAS form an unstructured domain that is positively charged and refractory to crystallization. Interestingly, this N terminus of cGAS was recently demonstrated to be essential for its activation by DNA through a process of phase condensation that assembles cGAS on long double-stranded DNA (*Du and Chen, 2018*). We reconstituted cGAS-deficient HeLa cells with GFP-cGAS fusions of full-length human cGAS and several truncation mutants corresponding to the structural domains of cGAS. To do this, we cloned a GFP-cGAS expression construct into a doxycycline-

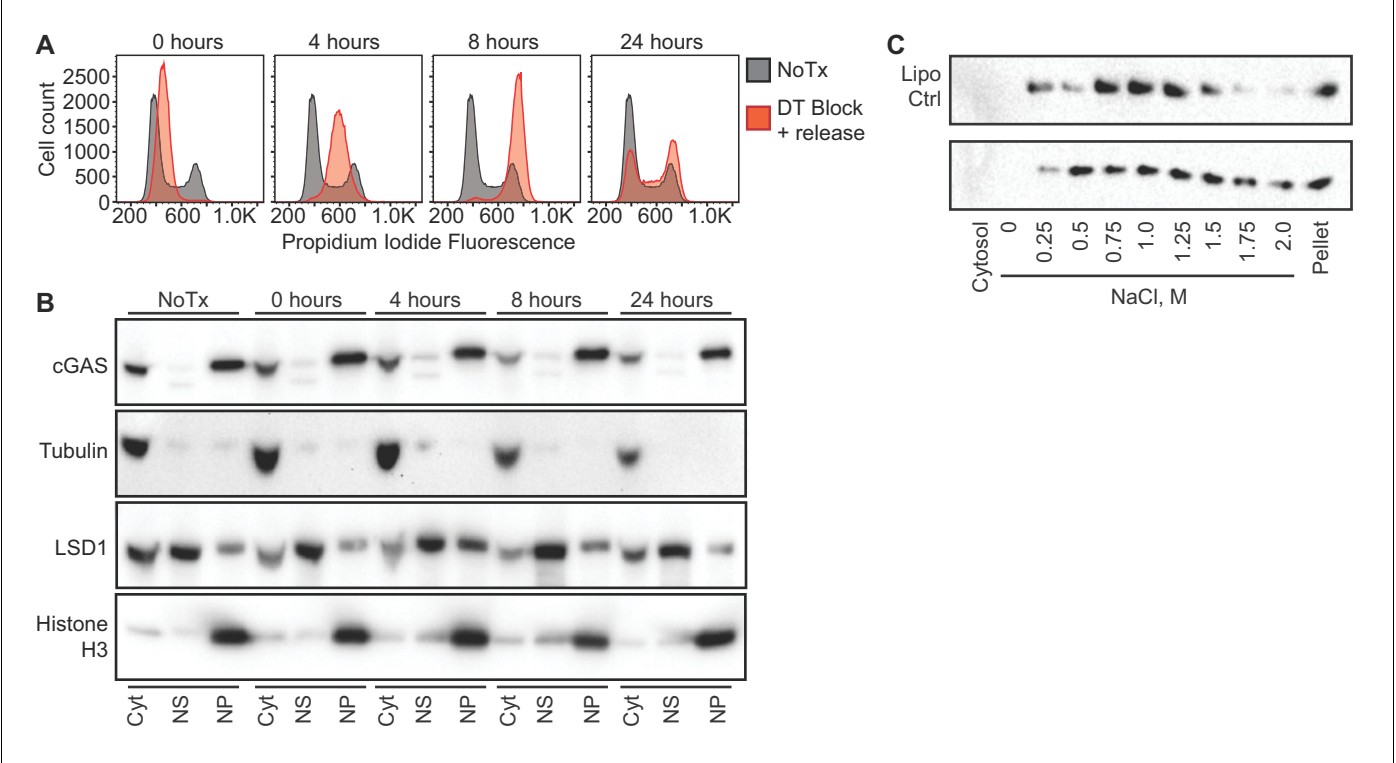

**Figure 3.** cGAS nuclear localization is independent of cell cycle phase or activation status. (**A**) HeLa cells were arrested at the G1/S border using double thymidine (DT) block, followed by release and harvest at the indicated time points for measurement of DNA content. (**B**) Cells from (**A**) were fractionated and cGAS localization was determined by western blot. (**C**) HeLa cells were transfected with Lipofectamine alone (Lipo) or with CT-DNA for 4 hr, followed by extraction, salt elution, and western blot for endogenous cGAS.

regulated lentiviral vector that enabled transduction of cGAS-deficient HeLa cells followed by selection for these transduced cells in the absence of cGAS expression. Induction of GFP-cGAS expression with doxycycline (Dox) allowed us to examine its localization in the absence of exogenous DNA stimulation, unlike standard transient transfections in which the plasmid DNA encoding cGAS also serves as a potent activating ligand. Our panel of truncation mutants revealed a number of important features of its nuclear tethering. First, we found that the GFP-cGAS (161-522) truncation mutant lacking the N terminus remained tethered in the nuclear pellet, and that the isolated N terminus of cGAS (1-161) localized to the cytosol and nuclear supernatant, with very little signal in the nuclear pellet (*Figure 4B*). Second, we found that removal of either the alpha-helical spine (161-213) or the C-terminal lobe of cGAS (383-522) resulted in a protein that was mislocalized and, in the case of the 213–522 mutant, also unstable (*Figure 4B*). Thus, the intact core of cGAS is required for its nuclear tethering.

We compared the full-length and N-terminal deletion mutant of cGAS in more detail. We tested a 100-fold range of dox concentrations that induced varying levels of the GFP-cGAS fusion constructs, from robust to nearly undetectable (*Figure 4C*). Consistent with the recent definition of the requirement of the N terminus for cGAS condensation onto DNA (*Du and Chen, 2018*), we found that the mutant lacking the N terminus of cGAS was severely compromised for DNA-activated cGAMP production at all levels of expression when compared to full-length cGAS (*Figure 4D*). However, sequential salt elution of the two forms of cGAS revealed nearly identical distribution and similarly tight tethering in the nucleus (*Figure 4E*).

We analyzed six additional mutants in the context of full-length murine cGAS to further explore the relationships between nuclear localization, nuclear tethering, DNA binding, DNA-mediated dimerization/oligomerization, and catalytic activity. The K335E mutant is severely defective for DNA binding and DNA-dependent activation (*Li et al., 2013*). The K395M/K399M mutation corresponds to the K407A/K411A mutation in the DNA-binding platform of human cGAS that results in reduced

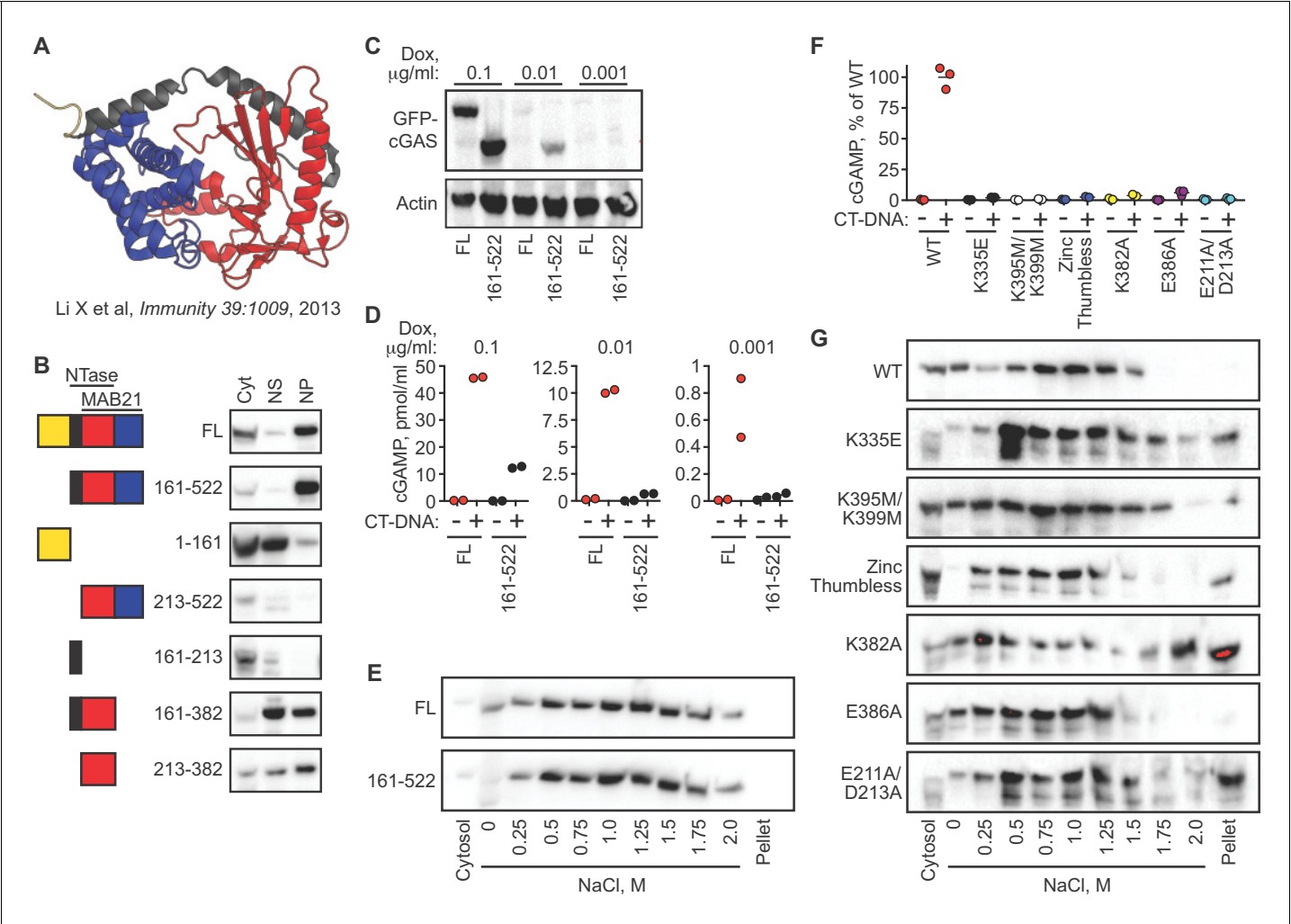

**Figure 4.** cGAS nuclear localization and tethering are independent of robust DNA binding, dimerization, condensation, and catalytic activity. (**A**) Structure of human cGAS with domains colorized. (**B**) TERT-immortalized human foreskin fibroblasts were reconstituted with the indicated Dox-inducible GFP-cGAS lentivirus constructs, treated with 0.1 µg/ml Dox for 24 hr, and then separated into cytosolic (Cyt), nuclear supernatant (NS), and nuclear pellet (NP) fractions. FL: Full-Length. (**C**) cGAS-deficient HeLa cells reconstituted with the indicated GFP-cGAS constructs were induced for 24 hr with three doses of Dox. Whole cell lysates that recover all cGAS were prepared and blotted with anti-GFP antibody. (**D**) Cells from (**C**) were transfected with CT-DNA for four hours, followed by cGAMP measurement in lysates by modified ELISA. (**E**) Cells described in C-D were treated with 0.1 µg/ml Dox for 24 hr to induce GFP-cGAS expression, then harvested and used for sequential fractionation and salt elution as in *Figure 2*. (**F**) Dox-inducible, full-length mouse cGAS constructs were introduced into hTERT-immortalized human fibroblasts. Cells were treated with 0.1 µg/ml Dox for 24 hr followed by stimulation for 4 hr and measurement of cGAMP in cell lysates. (**G**) Unstimulated cells from (**F**) were fractionated and blotted for cGAS. The online version of this article includes the following figure supplement(s) for figure 4:

**Figure supplement 1.** NONO and IFI16 are dispensable for cGAS nuclear localization and tethering.

DNA binding and defective DNA-mediated activation (*Civril et al., 2013*). The 'zinc thumb-less' mutant lacks amino acids 378–393 in mouse cGAS (390–405 in human), which form a protrusion that interacts with double-stranded DNA and is essential for cGAS activation (*Civril et al., 2013*). The K382A mutation results in decreased DNA binding and defective DNA-mediated dimerization, whereas the E386A mutation binds to DNA but fails to dimerize (*Li et al., 2013*). Finally, the E211A/ D213A mutant disrupts cGAS catalytic activity (*Li et al., 2013*). We cloned each of these cGAS constructs into the doxycycline-controlled lentiviral vector, introduced them into TERT-immortalized human fibroblasts, and then induced expression with doxycycline. We confirmed prior studies demonstrating that all six mutants are severely impaired for DNA-activated cGAMP production (*Figure 4F*; *Civril et al., 2013*; *Li et al., 2013*). Importantly, all six of these mutants remained

predominantly nuclear and tethered (*Figure 4G*). Together, our data reveal two important points about the requirements for cGAS localization versus its activation. First, the nuclear tethering of cGAS can be uncoupled from robust DNA binding, from DNA-activated dimerization, from DNA-activated condensation, and from catalytic activity, revealing separate mechanistic processes that govern the resting and activated states of cGAS. Second, it has been argued that our conditions of cGAS extraction and salt elution might result in the unnatural oligomerization and condensation of cytosolic cGAS onto DNA that might be liberated during the extraction, which could then cause such condensed cGAS to co-sediment with nuclei during the low speed centrifugation step (*Barnett et al., 2019*). Our findings argue against this possibility because the domains and specific amino acids that are essential for DNA-induced condensation are all dispensable for its nuclear localization and tethering.

We tested two additional potential cGAS tethering mechanisms. The nuclear protein NONO was recently found to bind to HIV capsid and mediate cGAS detection of HIV cDNA in the nucleus of human dendritic cells (*Lahaye et al., 2018*). Interestingly, NONO was responsible for the nuclear localization of a pool of cGAS that was extractable by ~400 mM salt in these cells (*Lahaye et al., 2018*). To test whether NONO is also essential for the tight tethering of the majority of nuclear cGAS, we generated four independent clonal lines of NONO-deficient HeLa cells (*Figure 4—figure supplement 1A*). We found that NONO-deficient cells produced normal amounts of cGAMP after DNA transfection (*Figure 4—figure supplement 1B*), consistent with the prior report (*Lahaye et al., 2018*). However, the tight nuclear tethering of cGAS was unaffected in NONO-deficient HeLa cells (*Figure 4—figure supplement 1A*). NONO may act as a 'bridge' to enable cGAS detection of virus-encapsidated DNA, as demonstrated in the prior study (*Lahaye et al., 2018*), but it is not the primary tether of cGAS. Lastly, we used a validated lentiCRISPR approach to disrupt the gene encoding IFI16 (*Gray et al., 2016*), which has been proposed to interact with cGAS and contribute to its activation (*Orzalli et al., 2015*). We found that IFI16 was extracted by ~420 mM salt into the nuclear supernatant and that IFI16 disruption resulted in no change in cGAS protein levels or its tight tethering in the nuclear pellet (*Figure 4—figure supplement 1C*).

## Intact chromatin is required for cGAS tethering

cGAS binds to DNA in a sequence-independent fashion, and its association with chromatin is thought to be generic, limited to mitosis, and mediated by its intrinsic affinity for DNA. To broadly assess the requirement for chromatin in the tethering of nuclear cGAS, we treated nuclei from both THP1 and HeLa cells with two broad-spectrum nucleases that digest both DNA and RNA. First, we digested the nuclear extracts after the zero salt lysis step for 30 min at 37° C with micrococcal nuclease, which was sufficient to convert the vast majority of chromatin into nucleosome-protected DNA fragments under 200 bp in length (*Figure 5A*). Second, we added salt-active nuclease (SAN) to the 250 mM salt elution step, which eliminated all detectable DNA from the samples (*Figure 5A*). In both cases, the nuclease digestions resulted in a collapsed salt elution profile for histones and cGAS, with the majority of cGAS released from the pellets at 250 mM and 500 mM salt (*Figure 5B*). These data demonstrate that intact chromatin is required for the organization of cGAS nuclear tethering.

## Identification of the evolutionarily conserved cGAS tethering surface

In the course of our studies of the relationship between DNA binding and nuclear tethering, we generated the R222E mutant of mouse cGAS (*Li et al., 2013*). This mutant, purified in recombinant form lacking the cGAS N terminus, was previously shown to be defective for binding to short (45 bp) double-stranded DNA oligonucleotides in vitro, with reduced DNA-activated cGAMP production (*Li et al., 2013*). However, in a transient transfection-based IFN-luciferase assay, R222E cGAS activated STING at levels comparable to WT cGAS (*Li et al., 2013*). We introduced the R222E mutation into full-length murine cGAS and transduced cGAS-deficient HeLa cells using our dox-inducible lentivirus system. Remarkably, we found that dox-induced expression alone, which did not activate WT cGAS, resulted in massive cGAMP production by the R222E mutant (*Figure 6A*). This constitutive cGAMP production by R222E cGAS was similar to the amount made by WT cGAS upon DNA transfection (*Figure 6A*). Importantly, DNA transfection did not further activate R222E, demonstrating that it was maximally active in the absence of exogenous DNA (*Figure 6A*).

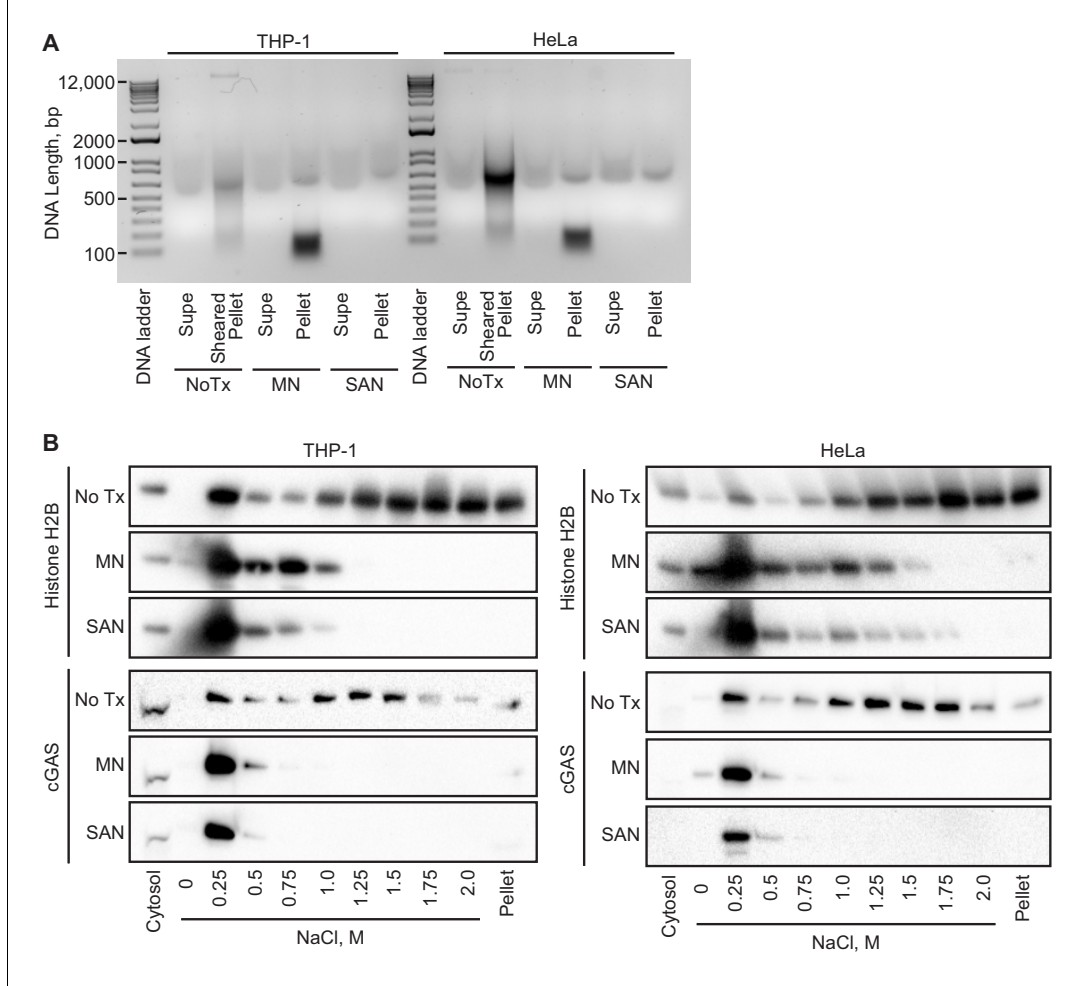

**Figure 5.** Intact chromatin is required for cGAS tethering. (**A**) THP-1 or HeLa cell nuclear extracts were left untreated (NoTx), treated after the 0 salt wash step with micrococcal nuclease (MN), or treated with at the 0.25 M NaCl elution step with Salt Active Nuclease (SAN). Supernatants (supe) and pellets were collected, and the untreated pellet was sonicated to shear large genomic DNA. DNA was extracted, run on an agarose gel, and visualized with SYBR Safe. (**B**) Extracts treated as described above were used for sequential salt elution, followed by blotting for Histone H2B or cGAS.

One potential explanation for this striking result is that the R222E mutation changed the conformation of the cGAS active site such that it was no longer dependent on the DNA-triggered structural rearrangements that are normally required for activation of cGAMP production. To test this directly, we made compound mutants of R222E with three of the mutants described in *Figure 4* to test the contributions of DNA binding (K335E; K395M/K399M) and DNA-activated dimerization (K382A). We found that all three of these compound mutants were completely defective for cGAMP production, in the absence or presence of exogenous DNA (*Figure 6A*). Thus, the massive constitutive activation of the cGAS R222E mutant is DNA-dependent.

We evaluated the salt elution profiles of the murine R222E cGAS mutant in both cGAS KO HeLa cells and TERT-HFFs. In both cases, we found that the R222E mutant remained predominantly nuclear, but it eluted at significantly lower salt concentrations (*Figure 6B*), similar to the elution profile of cGAS after digestion of nuclear extracts with nucleases (*Figure 5*). Taken together, these data demonstrate that the cGAS R222E mutant is untethered and constitutively active against self-DNA.

We mapped the location of murine cGAS R222 onto the crystal structure of DNA-bound murine cGAS (*Figure 6C*) (*Andreeva et al., 2017*), compared it to related proteins, and noted several important features for further study. R222 is found within the core NTase domain of cGAS, distant from the active site and located proximal to DNA in the structure, consistent with its original description as a DNA-binding residue (*Figure 6C*). We found that R222 and its surrounding surface define a

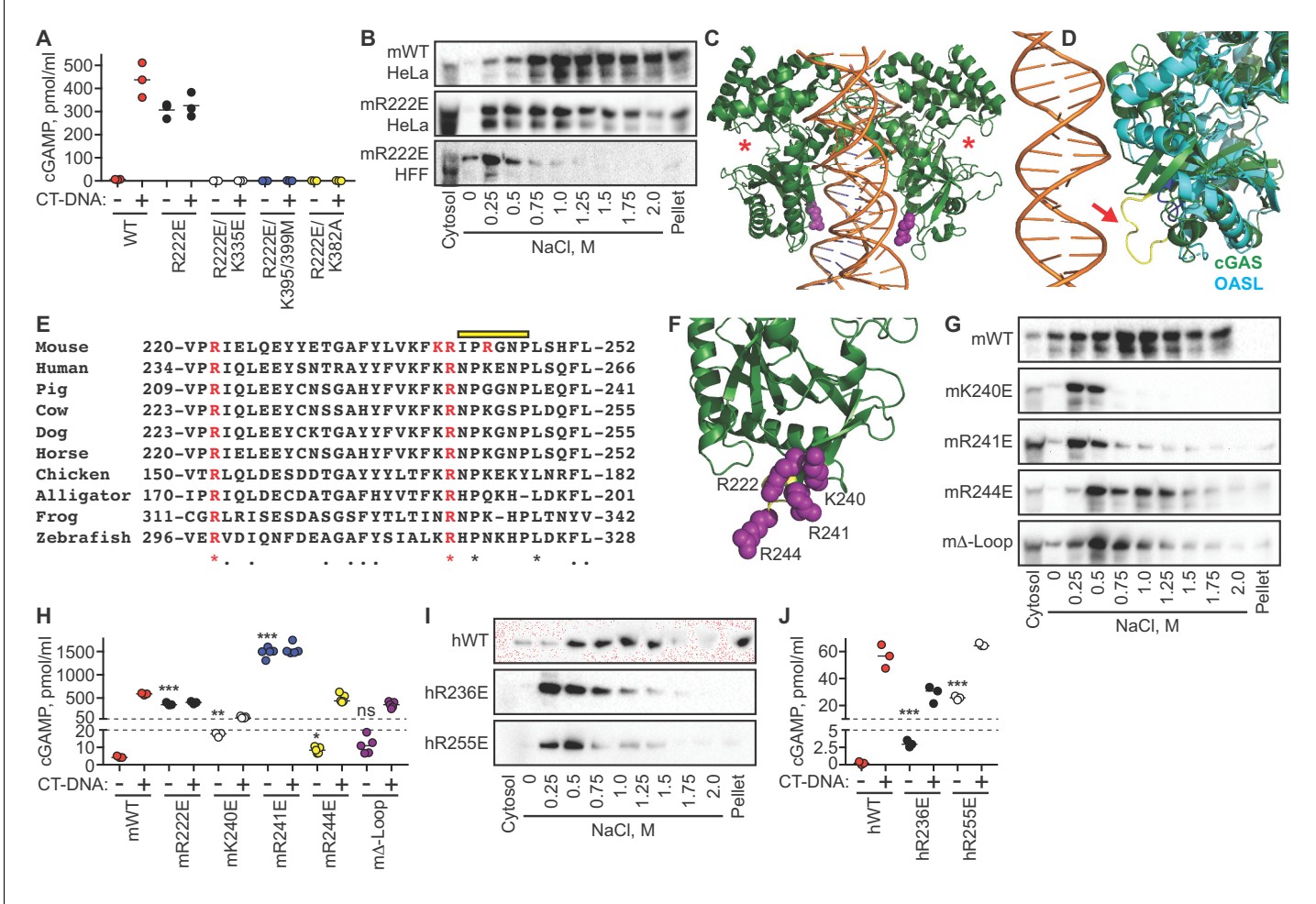

**Figure 6.** The cGAS tethering surface prevents autoreactivity. (**A**) cGAS KO HeLa cells were reconstituted with the indicated Dox-inducible murine cGAS lentivirus constructs, treated with 0.1 μg /ml Dox for 24 hr, then transfected for four hours with lipofectamine alone or lipofectamine + CT DNA, followed by measurement of cGAMP in cell lysates. (**B**) Salt elution profiles of WT and R222E mouse cGAS; the top two panels are in HeLa cells and the bottom panel is in TERT-immortalized human fibroblasts. (**C**) Crystal structures of mouse cGAS (green) assembled on DNA, modeled from PDBID 5N6I. DNA is orange, R222 is highlighted in purple spheres, and the locations of the active sites are noted with red asterisks. (**D**) Overlay of mouse cGAS (green) and human OASL (cyan; PDBID 4XQ7). The unique loop in cGAS is colored yellow and indicated with the red arrow. (**E**) Alignments of cGAS across vertebrate phylogeny, with conserved, positively charged residues highlighted in red and the central loop amino acids indicated by the yellow bar. (**F**) The positively charged residues near R222 are highlighted in purple spheres. (**G**) Salt elution profiles of cGAS KO HeLa cells transduced with the indicated murine cGAS constructs after induction with 0.1 μg/ml Dox for 24 hr. (**H**) Cells from (**G**) were treated with 0.1 μg/ml Dox for 24 hr, then transfected for four hours with lipofectamine alone or lipofectamine + CT-DNA, followed by measurement of cGAMP in cell lysates. (**I**) Salt elution profiles of cGAS KO HeLa cells transduced with the indicated human cGAS constructs after induction with 0.1 μg/ml Dox for 24 hr. (**J**) Cells from (**I**) were treated as in (**H**), followed by measurement of cGAMP in cell lysates. Statistical comparisons were made between resting WT cGAS and each mutant, using one-way ANOVA of log-transformed biological replicates. *p=0.0155; **p=0.0018; ***p<0.0001; ns: not significant.

module that is unique to cGAS and not conserved in OAS proteins (*Figure 6D*; *Ibsen et al., 2015*). This module includes a protruding loop that is poorly resolved among numerous cGAS structures, suggesting dynamic flexibility (*Figure 6D*). Phylogenetic analysis of this surface reveals that R222 and R241 of murine cGAS are completely conserved across ~360 million years of vertebrate evolution, despite poor overall sequence conservation in this region of the protein (*Figure 6E*). Notably, R241, which is immediately adjacent to R222 in the cGAS structure, projects away from DNA and does not participate in DNA binding (*Figure 6F*). Finally, we found two additional positively charged residues within this surface that are largely - but not completely - conserved, corresponding to K240 and R244 of murine cGAS (*Figure 6E–F*).

We generated the K240E, R241E, and R244E single point mutants of murine cGAS, along with a 'Δ-Loop' mutant in which the central amino acids of the protruding loop (IPRGNP) were replaced with a flexible SGSGSG sequence. We evaluated the effects of these mutations on cGAS nuclear tethering and on constitutive and DNA-activated cGAMP production. We found that the K240E mutant was untethered and significantly constitutively active upon expression alone (*Figure 6G–H*), but it was also impaired by 90% for DNA-activated cGAMP production compared to WT cGAS, suggesting that K240 plays a dual role in both nuclear tethering and in DNA-dependent activation of cGAS. Remarkably, the R241E mutant was untethered, even more constitutively active than R222E, and not further activated by DNA transfection (*Figure 6G–H*). The amount of cGAMP produced constitutively by R241E was 300 times more than resting WT cGAS and nearly triple the amount produced by DNA-activated WT cGAS. Finally, the R244E mutant and the Δ-Loop mutant were mildly less tethered and mildly constitutively active, but they remained strongly inducible by DNA transfection (*Figure 6G–H*). Thus, specific amino acids are essential for both tight tethering of nuclear cGAS and for prevention of cGAS activation by self-DNA.

To test whether tethering and negative regulation of cGAS are evolutionarily conserved, we generated mutants in human cGAS (hcGAS) corresponding to murine R222 (hR236E) and murine R241 (hR255E). We found that both hR236E cGAS and R255E hcGAS mutants were untethered and constitutively active, with R255E hcGAS constitutively producing over 100 times more cGAMP than WT hcGAS (*Figure 6I–J*). The amounts of constitutive cGAMP produced by the human cGAS mutants were lower than the amounts made by the corresponding murine mutants, likely reflecting the fact that the human cGAS enzyme is intrinsically less active than murine cGAS (*Zhou et al., 2018*). Moreover, these mutants remained further inducible by DNA (*Figure 6I–J*), suggesting subtle differences in the contributions of these specific amino acids to regulation of mouse and human cGAS. Taken together, these findings identify the tethering surface of cGAS and demonstrate that tight nuclear tethering is a fundamental and evolutionarily conserved mechanism that prevents cGAS activation by self-DNA.

## Discussion

Put simply, the cytosolic DNA sensing model holds that cGAS is a cytosolic protein that is activated by binding to double-stranded DNA that appears in the cytosol (*Stetson and Medzhitov, 2006*). Our findings demonstrate that DNA detection by cGAS is more complex than this simple model, and they warrant further study of the relationship between resting and activated cGAS, together with the spatial and biochemical transitions that accompany its activation.

We show, using microscopy and biochemical fractionation, that the great majority of endogenous cGAS is nuclear prior to its activation, in all cells tested, in all phases of the cell cycle. Many recent studies in cell lines, most using overexpressed, tagged cGAS, have shown images of cGAS localized to the cytosol and absent from the nucleus, although nuclear cGAS has been observed in mouse fibroblasts expressing GFP-cGAS (*Yang et al., 2017*) and in mouse hematopoietic stem cells (*Xia et al., 2018*). Others have recently reported regulated translocation of a fraction of overexpressed cGAS from the cytosol to the nucleus in response to DNA damage (*Liu et al., 2018*). Finally, studies of cGAS localization to micronuclei envision the recruitment of cytosolic cGAS to cytosolic micronuclei upon rupture of the membrane surrounding these structures (*Bartsch et al., 2017*; *Dou et al., 2017*; *Glück et al., 2017*; *Harding et al., 2017*; *Mackenzie et al., 2017*; *Yang et al., 2017*). We cannot explain these disparate findings, but our data demonstrate that endogenous cGAS is in the nucleus prior to its activation, in the absence of exogenous DNA damaging agents, and prior to the appearance of micronuclei that require mitosis for their formation.

We find that endogenous cGAS is tethered tightly in the nucleus by a force that is remarkably resistant to salt extraction, which explains why this pool of cGAS has been missed when conventional cytosolic and nuclear extracts have been reported in prior studies. We demonstrate that cGAS nuclear localization and nuclear tethering do not require the specific amino acids in cGAS that are essential for robust DNA binding, for DNA-induced oligomerization, for DNA-induced condensation into phase-separated liquid droplets, or for catalytic activity. We show that cGAS nuclear tethering requires intact chromatin. Finally, we identify the tethering surface of cGAS, which resides within NTase domain, and we show that failure to tether cGAS results in its constitutive activation by self-DNA.

By identifying and mutating the tethering surface of cGAS, we uncover a number of important features that govern cGAS localization and its disposition prior to activation. First, the location of the tethering surface corroborates the domain mapping and mutational analysis presented in *Figure 4*, and it demonstrates that cGAS nuclear tethering and DNA-based activation are two distinct processes. Second, our findings are incompatible with a number of recent studies that have suggested an important role for the N terminus of cGAS in localization to either the cytosolic plasma membrane (*Barnett et al., 2019*) or to the nucleus (*Gentili et al., 2019*), or a primary role for cGAS DNA binding in its association with chromatin (*Jiang et al., 2019*). In particular, our data offer an alternative explanation for the observation that cGAS co-sediments with plasma membranes upon fractionation (Figure 1 of *Barnett et al., 2019*): in order to reveal this pool of cGAS, cellular extracts needed to be treated prior to fractionation with nuclease, which liberates tethered nuclear cGAS from chromatin (*Figure 5*).

What is the nature of the tether that tightly binds cGAS to chromatin? A recent study found that recombinant, truncated cGAS lacking its N terminus binds more tightly to purified, recombinant nucleosomes than to naked DNA (*Zierhut et al., 2019*), which is consistent with the requirement for chromatin in cGAS nuclear tethering (*Figure 5*). However, the relevance of these in vitro studies to the tight nuclear tethering of cGAS that we define in cells remains unclear. Interestingly, the residues that are important for nuclear tethering overlap with, but are distinct from, one of the DNA-binding surfaces of cGAS (*Figure 6*). Because of this, we propose that interaction between cGAS and the tethering factor(s) would not be compatible with assembly of cGAS on DNA, which would explain why resting nuclear cGAS is not activated by the billions of base pairs of genomic DNA that reside in the same compartment. If cGAS were tethered to nuclear chromatin through DNA binding, it would be difficult to explain how it is not constitutively activated. Moreover, if resting cGAS were saturated with self-DNA (in an undefined manner that would not activate cGAS), then its activation would require the titration of cGAS off 'inert' self-DNA and onto 'activating' foreign DNA. How could DNA tether resting cGAS and then be replaced with the identical chemical structure to activate it? There is no biological precedent for such a model, and the fact that loss of tethering results in massive cGAS activation by self-DNA strongly suggests that tethered nuclear cGAS is physically sequestered from DNA. Based on these considerations, we speculate that tethered nuclear cGAS does not directly interact with genomic DNA, and that the tethering mechanism maintains resting cGAS in a state that is competent to detect foreign DNA. More broadly, we propose that analogous tethers might serve as platforms to regulate transactions between other DNA-binding proteins and DNA in the nucleus, serving as a mechanism to prevent inappropriate activation of enzymes that require and/or act on DNA.

We show that untethered cGAS is constitutively active against self-DNA. Thus, cGAS is not 'inert' prior to its encounter with foreign DNA, as has been envisioned in the cytosolic DNA sensing model. Instead, tight nuclear tethering is an active process that is essential for preventing autoreactivity, a finding that has a number of implications. First, there must be a regulated step prior to assembly of cGAS onto DNA that allows its activation. We propose a model of 'regulated desequestration' of nuclear cGAS and DNA that is required for its full activation, which is then followed by the phase separation and condensation of cGAS onto DNA that was recently demonstrated (*Du and Chen, 2018*). Our model explains the fact that resting cGAS is not active, and it also allows for nuclear cGAS, together with the factors that regulate tethering, to distinguish self DNA from foreign DNA within the same compartment. We note that the production of the diffusible second messenger cGAMP by cGAS provides a simple explanation for how activated nuclear cGAS could trigger the cytosolic signaling complex of STING, TBK1, and IRF3. Second, there may be conditions in which perturbation of nuclear tethering alone would result in activation of cGAS by self-DNA. Such conditions might include genetic polymorphisms in cGAS or the tether that govern the strength of cGAS tethering, or environmental changes that relax cGAS tethering in response to stress or damage. In these cases, cGAS autoreactivity would not require the production of a distinct DNA ligand that is erroneously sensed as 'foreign,' nor would it require that such DNA leave the nucleus. Finally, because untethering alone is sufficient to activate cGAS without the need for foreign DNA, it is possible that self-DNA contributes to ligand-dependent cGAS activation in the context of viral infection. In other words, upon 'regulated desequestration,' cGAS assembly on either self-DNA or foreign DNA would achieve the identical result of cGAMP production and antiviral response.

In conclusion, we have found that two separate processes govern the resting and activated states of cGAS, and we propose that a deeper understanding of these states will illuminate new aspects of cGAS biology, with implications for mechanisms of self/non-self discrimination by the innate immune system.

# Materials and methods

**Key resources table**

| Reagent type (species) or resource | Designation | Source or reference | Identifiers | Additional information |
|---|---|---|---|---|
| Cell line (human) | HeLa | ATCC | CCL-2 | |
| Cell line (human) | SiHa | ATCC | HTB-35 | |
| Cell line (human) | THP1 | ATCC | TIB-202 | |
| Cell line (human) | HEK 293T | ATCC | CRL-3216 | |
| Cell line (human) | Primary human foreskin fibroblasts | Millipore | Cat # SCC-058 | |
| Cell line (mouse) | mouse bone marrow macrophages, C57BL/6J | Gray EE et al, J Immunol 2015 195:1939 PMID: 27496731 | | |
| Cell line (human) | HeLa H1 LentiCRISPR control clonal lines | This Paper | | |
| Cell line (human) | HeLa cGAS LentiCRISPR clonal lines | This Paper | | |
| Cell line (human) | Hela cGAS KO pSLIK GFP | This Paper | | |
| Cell line (human) | Hela cGAS KO pSLIK GFP FL mcGAS | This Paper | | |
| Cell line (human) | HeLa cGAS KO pSLIK GFP mcGAS 161–522 | This Paper | | |
| Cell line (human) | HeLa cGAS KO pSLIK mcGAS WT | This Paper | | |
| Cell line (human) | HeLa cGAS KO pSLIK mcGAS R222E | This Paper | | |
| Cell line (human) | HeLa cGAS KO pSLIK mcGAS R222E/K335E | This Paper | | |
| Cell line (human) | HeLa cGAS KO pSLIK mcGAS R222E/K335E | This Paper | | |
| Cell line (human) | HeLa cGAS KO pSLIK mcGAS R222E/K395M/K399M | This Paper | | |
| Cell line (human) | HeLa cGAS KO pSLIK mcGAS R222E/K382A | This Paper | | |
| Cell line (human) | HeLa cGAS KO pSLIK mcGAS K240E | This Paper | | |
| Cell line (human) | HeLa cGAS KO pSLIK mcGAS R241E | This Paper | | |
| Cell line (human) | HeLa cGAS KO pSLIK mcGAS R244E | This Paper | | |
| Cell line (human) | HeLa cGAS KO pSLIK mcGAS ΔLoop | This Paper | | |
| Cell line (human) | HeLa cGAS KO pSLIK hcGAS WT | This Paper | | |

*Continued on next page*

*Continued*

| Reagent type (species) or resource | Designation | Source or reference | Identifiers | Additional information |
|---|---|---|---|---|
| Cell line (human) | HeLa cGAS KO pSLIK hcGAS R236E | This Paper | | |
| Cell line (human) | HeLa cGAS KO pSLIK hcGAS R255E | This Paper | | |
| Cell line (human) | hTERT human fibroblast pSLIK GFP-mcGAS full length (FL) | This Paper | | |
| Cell line (human) | hTERT human fibroblast pSLIK GFP-mcGAS (161-522) | This Paper | | |
| Cell line (human) | hTERT human fibroblast pSLIK GFP-mcGAS (1-161) | This Paper | | |
| Cell line (human) | hTERT human fibroblast pSLIK GFP-mcGAS (213-522) | This Paper | | |
| Cell line (human) | hTERT human fibroblast pSLIK GFP-mcGAS (161-213) | This Paper | | |
| Cell line (human) | hTERT human fibroblast pSLIK GFP-mcGAS (161-382) | This Paper | | |
| Cell line (human) | hTERT human fibroblast pSLIK GFP-mcGAS (213-382) | This Paper | | |
| Cell line (human) | hTERT human fibroblast pSLIK mcGAS K335E | This Paper | | |
| Cell line (human) | hTERT human fibroblast pSLIK mcGAS K395M/K399M | This Paper | | |
| Cell line (human) | hTERT human fibroblast pSLIK mcGAS Zinc thumbless | This Paper | | |
| Cell line (human) | hTERT human fibroblast pSLIK mcGAS K382A | This Paper | | |
| Cell line (human) | hTERT human fibroblast pSLIK mcGAS E386A | This Paper | | |
| Cell line (human) | hTERT human fibroblast pSLIK mcGAS E211A/D213A | This Paper | | |
| Cell line (human) | hTERT human fibroblast H1 lentiCRISPR control | Gray EE et al, Immunity 2016 45:255 PMID: 27496731 | | |
| Cell line (human) | hTERT human fibroblast IFI16 lentiCRISPR | Gray EE et al, Immunity 2016 45:255 PMID: 27496731 | | |
| Cell line (mouse) | mouse bone marrow macrophages, *Cgas-/-* | Gray EE et al, J Immunol 2015 195:1939 PMID: 26223655 | | |
| Cell line (mouse) | WT MEF (mouse embryonic fibroblasts) | Gray EE et al, J Immunol 2015 195:1939 PMID: 26223655 | | |
| Transfected construct | pSLIK-Neo | Addgene | Plasmid #25735 | |

*Continued on next page*

*Continued*

| Reagent type (species) or resource | Designation | Source or reference | Identifiers | Additional information |
|---|---|---|---|---|
| Transfected construct | pSLIK-Blasticidin | This Paper | | |
| Antibody | rabbit monoclonal anti-cgas D1D3G | Cell Signaling | cat # 15102S | |
| Antibody | rabbit monoclonal anti-cgas, mouse specific | Cell Signaling | cat # 31659 | |
| Antibody | rabbit monoclonal anti-LSD1 C69G12 | Cell Signaling | cat # 2184S | |
| Antibody | rabbit polyclonal anti-histone H3 | abcam | cat # ab1791 | |
| Antibody | rabbit polyclonal anti-histone H2B | abcam | cat # ab1790 | |
| Antibody | rabbit polyclonal anti-GFP | abcam | cat # 6556 | |
| Antibody | mouse monoclonal anti-Ifi16 | abcam | cat # 55328 | |
| Antibody | rabbit polyclonal anti-NONO | Sigma Aldrich | cat # N8789-200UL | |
| Antibody | rabbit polyclonal anti- $\alpha/\beta$ tubulin | Cell Signaling | cat # 2148 | |
| Antibody | mouse monoclonal anti-$\beta$ actin | SIGMA | cat # A5441 | |
| Antibody | rabbit IgG control | Fisher | cat # 10500C | |
| Antibody | rabbit polyclonal anti-cgas | SIGMA | cat # HPA031700 | |
| Antibody | mouse monoclonal anti-$\beta$-tubulin | Cell Signaling | cat # 86298 | |
| Commercial assay or kit | NE-PER nuclear and cytoplasmic extraction reagents | Thermo Fisher Scientific | cat # PI78835 | |
| Commercial assay or kit | propidium iodide | SIGMA | cat # P4170 | |
| Commercial assay or kit | Luciferase Assay System | Promega | cat # E4550 | |
| Commercial assay or kit | 2',3'-Cyclic GAMP Direct EIA Kit | Arbor Assays | cat # K067-H1 | |
| Chemical compound, drug | thymidine | VWR | cat # 80058–750 | |
| Chemical compound, drug | SAN (salt active nuclease) | SIGMA | cat # SRE0015 | |
| Chemical compound, drug | calf thymus DNA | SIGMA | cat # D4764 | |
| Chemical compound, drug | micrococcal nuclease | New England Biolabs | cat # M0247S | |
| Chemical compound, drug | DAPI | SIGMA | cat # D9542 | |
| Chemical compound, drug | SYBR safe | Apex Bio | cat # A8743 | |
| Chemical compound, drug | DRAQ5 | Thermo Fisher | cat # 62251 | |
| Chemical compound, drug | NP-40 substitute | SIGMA | cat # 74385 | |

*Continued on next page*

*Continued*

| Reagent type (species) or resource | Designation | Source or reference | Identifiers | Additional information |
|---|---|---|---|---|
| Chemical compound, drug | protease inhibitor tablet | Pierce | cat # PIA32955 | |
| Chemical compound, drug | Prolong Gold Antifade mountant | Thermo Fisher | cat # P36930 | |
| Chemical compound, drug | Roche Block | Sigma | cat # 11921673001 | |
| Other | Immobilon-FL PVDF, 0.45 µm western blot membrane | SIGMA | cat # IPFL00010 | |
| Other | Immobilon-PSQ PVDF, 0.2 µm western blot membrane | SIGMA | cat # ISEQ00010 | |

## Cell lines and mice

The following human cell lines were purchased from ATCC. Some of these lines are on the list of commonly misidentified cell lines maintained by the International Cell Line Authentication Committee; we used STR profiling from the University of Arizona Genetics Core to confirm their identity. We also tested for Mycoplasma contamination using a commercially available kit (ABM, cat # G238). All cell lines used in this study tested negative for Mycoplasma contamination.

HeLa: ATCC CCL-2
SiHa: ATCC HTB-35
THP1: ATCC TIB-202

Primary human foreskin fibroblasts were purchased from Millipore, cat # SCC058, and immortalized with retrovirus expressing h-TERT for this study.

C57BL/6J mice were purchased from Jackson Laboratories. *Cgas-/-* mice were generated previously, and bone marrow macrophages and MEFs were made as described (*Gray et al., 2015*).

## Immunofluorescence microscopy

HeLa cells or primary mouse bone marrow-derived macrophages (BMMs) were seeded onto glass coverslips overnight, then fixed and permeabilized with ice cold methanol at −20 degrees C for 10 min. Cells were then washed in PBS and blocked at room temperature for 2 hr (HeLa cells in Roche block in PBS; BMM in 5% normal goat serum in PBS). Cells were then incubated with primary antibody in block overnight at 4° C. For HeLa cells, we used anti-cGAS CST D1D3G Ab at 1:50 and anti-β-tubulin CST D3U1W Ab at 1:100. For mouse BMMs, we used anti-cGAS CST D3080 Ab at 1:250. Cells were washed in PBS and incubated with secondary Ab (goat anti-rabbit Alexafluor 488, goat anti-mouse Alexafluor 546, Invitrogen) at 1:500 for 1 hr at room temperature. Cells were then washed with PBS, stained with DAPI and mounted on glass slides with ProLong Gold Antifade Mountant (Thermo Fisher). For PFA fixation, cells were fixed in 4% PFA for 10 min, washed in PBS, then permeabilized in 0.1% Triton X-100 in PBS for 10 min. Cells were then washed in PBS, blocked for one hour at room temperature in Roche block in PBS, then stained as stated above. Images were captured with a Nikon C2RSi Scanning Laser Microscope, using a Plan Apo VC 60 × Oil DIC N2 objective in the 405, 488, and 562 dichroic channels and z-steps of 0.125 µm with NISElements software, and then pseudocolored using Fiji open source software. Z-series and 3D volume views were created with NISElements software.

## Generation of cGAS and NONO knockout HeLa cells

LentiCRISPR vector generation and lentiviral transductions were done as described previously (*Gray et al., 2016*). Clonal lines of HeLa cells were generated by limiting dilution and then assessed for targeting by Sanger sequencing, western blot analysis, and functional assays for cGAMP production. The guide RNAs used were:

H1 off-target control: 5'-(G)ACGGAGGCTAAGCGTCGCAA (*Sanjana et al., 2014*), where the (G) denotes a nucleotide added to enable robust transcription from the U6 promoter; cGAS: 5'-GGCGCCCCTGGCATTCCGTG<u>CGG</u>, where the underlined sequence denotes the Protospacer Adjacent Motif (PAM); NONO: 5'-CTGGACAATATGCCACTCCGT<u>GG</u>.

## Amnis imagestream analysis

cGAS KO HeLa cells transduced with pSLIK GFP-mcGAS were treated with 1 µg/ml dox for 24 hr. Cells were washed in PBS and then rested in complete media for 24 hr. Cells were then released from the plate with trypsin, washed in PBS, and stained with 3.125 µM DRAQ5 in PBS before running on an Amnis Imagestream X Mark II imaging cytometer. Data were analyzed with Ideas software (version 6.2).

## Salt extractions

We modified a published protocol for histone extraction (*Shechter et al., 2007*). Cells were pelleted, washed in PBS, resuspended in 1 mL extraction buffer (10 mM Hepes pH 7.9, 10 mM KCl, 1.5 mM MgCl$_2$, 0.34 M sucrose, 10% glycerol, 0.2% NP-40, and Pierce protease inhibitors), and incubated on ice for 10 min with occasional vortexing. Nuclei were spun at 6500 x g for 5 min at 4° C. The cytosolic fraction (supernatant) was collected for further analysis. Nuclei were then washed for 1 min on ice in extraction buffer without NP-40 and spun at 6500 x g for 5 min at 4°C. Pelleted nuclei were then resuspended in 1 mL zero salt buffer (3 mM EDTA, 0.2 mM EGTA, and protease inhibitors), and vortexed intermittently for 1 min (10 s on, 10 s off). Nuclei were then incubated on ice for 30 min, vortexing for 15 s every 10 min. Lysates were then spun at 6500 x g for 5 min at 4° C. The zero salt supernatant was collected for further analysis. The remaining pellets were then resuspended in first salt buffer (50 mM Tris-HCl, pH 8.0, 0.05% NP-40, 250 mM NaCl), incubated on ice for 15 min with vortexing for 15 s every 5 min. Lysates were spun at full speed (15,000 rpm) at 4°C for 5 min. Supernatants were collected for further analysis. Subsequent salt extractions were performed on the pellet with sequential increases in NaCl concentration (500 mM, 750 mM, 1 M, 1.25 M, 1.5 M, 1.75 M, and 2 M). Samples in each salt wash were incubated on ice for 15 min with vortexing for 15 s every 5 min. Supernatants following each salt condition were collected for further analysis. The final pellet was then resuspended in salt buffer with 2M NaCl and sonicated with a Covaris M220 focused ultrasonicator at 5% ChIP (factory setting), or digested with Salt Active Nuclease (SAN) where the buffer was supplemented with 20 mM MgCl$_2$. All samples were supplemented with denaturing SDS-PAGE sample buffer, separated on acrylamide gels, transferred to membranes for western blot (0.2 µM pore size for histone blots, 0.45 µM pore size for all other blots), and blotted with the indicated primary and secondary antibodies using standard approaches. Western blot images were acquired and densitometry analysis was performed using a BioRad Chemidoc and associated software.

## NE-PERS kit modification

The NE-PERS kit instructions (Thermo Fisher) were followed completely, with the following modification: after spinning the pellet out of the NER buffer, the supernatant was removed and saved as 'nuclear supernatant (NS)'. The remaining pellet was resuspended in a volume of NER buffer equal to the first, and either sonicated (using Covaris M220 5% ChIP factory setting), or digested with SAN in NER buffer supplemented with 20 mM MgCl$_2$. This was then saved as 'nuclear pellet (NP)'.

## Double Thymidine block

Cells were seeded onto plates to achieve 40% confluency. The next day cells were treated with 2 mM thymidine in complete media for 19 hr. Cells were then washed in warm PBS and rested in complete media for 9 hr. Cells were then treated again with 2 mM thymidine in complete media for 16 hr. Cells were then either harvested for analysis by western blot or flow cytometry, or washed and returned to complete media for harvest at post-release time points. Flow cytometry analysis was performed as described above.

## cGAMP quantitation assay

Cells were plated at 100,000 cells/well in a 24 well tissue culture dish. 24 hr later, cells were transfected with either 10 μg/ml CT-DNA in lipofectamine 2000 (Invitrogen; ratio of 1 μL lipofectamine per 1 μg CT-DNA) (Stetson and Medzhitov, 2006), or with an identical volume of lipofectamine 2000 alone. 4 hr later, cells were harvested and lysates were prepared using cGAMP EIA assay protocol provided by manufacturer (Arbor Assays), in a volume of 200 μL sample suspension buffer.

## Constructs

The pSLIK-Neo doxycycline-inducible lentiviral vector was obtained from Addgene and modified to replace the Neo cassette with a blasticidin resistance cassette. GFP fusions to the murine cGAS open reading frame were generated by PCR mutagenesis and designed to incorporate a four-glycine flexible linker between the last amino acid of GFP and the first amino acid of cGAS. Lentivirus production and blasticidin selection were done using standard techniques.

cGAS-deficient HeLa cells were reconstituted with Dox-inducible lentiviruses encoding GFP, the indicated GFP-human cGAS fusions, or the indicated GFP-mouse cGAS constructs. Cells were plated at 50,000 cells per well in a 24 well plate for quantitation of cGAMP, or 250,000 cells per well in a 6-well plate for salt extractions. 24 hr later, cells were treated with doxycycline for 24 hr. Then, cells were harvested directly for anti-GFP western blot from the 6-well dishes, and the 24-well dishes were transfected with either 10 μg/mL CT-DNA complexed with lipofectamine 2000, or with lipofectamine 2000 alone. 4 hr later, lysates were prepared and analyzed for cGAMP content as described above (whole cells lysed in 200 μL sample suspension buffer).

## Nuclease digestions and salt elutions

Salt extractions were performed as described above with the following modifications. $1 \times 10^6$ cells were used for each condition. Following the zero salt wash, all samples were resuspended in digestion buffer (50 mM Tris pH 8.0, 0.05% NP-40, 1 mM $MgCl_2$, 5 mM $CaCl_2$). For the MNase digestion, MNase was added at 20,000 gel units per sample. For the SAN digestion, 50 units SAN nuclease plus 20 mM $MgCl_2$ were added at the 250 mM NaCl step. All samples were incubated at 37°C for 10 min. Samples were then spun down and supernatants and pellets were separated and then processed for western blots using the salt elution protocol. For analysis of DNA content, the MNase-digested supernatants and pellets were analyzed after digestion and before commencement of salt elution. For the SAN digestion, the pellet was collected after the 500 mM NaCl elution for assessment of DNA content. DNA was run on an agarose gel, stained using SYBR-Safe reagent (Apex Bio), and visualized using a BioRad Chemidoc.

## Experimental replicates and reproducibility

All data presented in this paper are representative of 2–4 independent experiments with comparable results.

## Acknowledgements

We are grateful to Quinton Dowling, Neil King, and Emily Schutsky for their help with cGAS structural analysis, to Andrew Oberst and Naeha Subramanian for the pSLIK-Neo vector, to Michael Gale, Jr for use of the confocal microscope, and to the entire Stetson lab for helpful discussions.

## Additional information

### Funding

| Funder | Grant reference number | Author |
| --- | --- | --- |
| National Institutes of Health | AI084914 | Daniel B Stetson |
| Jane Coffin Childs Memorial Fund for Medical Research | | Hannah E Volkman |
| Burroughs Wellcome Fund | 1013540 | Daniel B Stetson |

| Howard Hughes Medical Institute | 55108572 | Daniel B Stetson |
| Bill and Melinda Gates Foundation | OPP1156262 | Daniel B Stetson |
| Cancer Research Institute | A84161 | Elizabeth E Gray |

The funders had no role in study design, data collection and interpretation, or the decision to submit the work for publication.

## Author contributions

Hannah E Volkman, Stephanie Cambier, Data curation, Formal analysis, Investigation, Methodology; Elizabeth E Gray, Investigation, Methodology; Daniel B Stetson, Conceptualization, Resources, Formal analysis, Supervision, Funding acquisition

## Author ORCIDs

Daniel B Stetson [iD] https://orcid.org/0000-0002-5936-1113

## Decision letter and Author response

Decision letter https://doi.org/10.7554/eLife.47491.sa1
Author response https://doi.org/10.7554/eLife.47491.sa2

# Additional files

## Supplementary files

• Transparent reporting form

## Data availability

All data generated or analyzed during this study are included in the manuscript and supporting files.

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
