## [Decision Letter]

[Editors’ note: a previous version of this study was rejected after peer review, but the authors submitted for reconsideration. The first decision letter after peer review is shown below.]

Thank you for submitting your work entitled "cGAS is predominantly a nuclear protein" for consideration by *eLife*. Your article has been reviewed by two peer reviewers, one of whom is a member of our Board of Reviewing Editors, and the evaluation has been overseen by a Senior Editor. The reviewers have opted to remain anonymous.

Our decision has been reached after consultation between the reviewers. Based on these discussions and the individual reviews below, we regret to inform you that your work will not be considered further for publication in *eLife*.

Although the reviewers felt the manuscript dealt with an issue that has not received much attention, i.e., the localization of cGAS to the nucleus, the consensus was that this issue had been previously described. As such, the observation is not entirely novel. Other key aspects of the current manuscript, including the functional impact of cGAS tethering to chromatin, remain unaddressed in a convincing manner. Other conclusions are also not clearly supported by the presented experimental data. Thus, the reviewers and editors decided that the paper is unacceptable for publication at *eLife* in its current form.

The complete reviews are given below, in hopes that you will find them useful for revising the manuscript for submission elsewhere. You will note that there was some disagreement on the novelty of the findings and other aspects of the paper, as is expected. However, the consensus statement above was produced after extensive discussions between the reviewers and the editors.

Reviewer #1:

This manuscript shows that endogenous cGAS, a nucleic acid sensor, is predominately localized to the nucleus. This observation suggests that current models of "cytosolic nucleic acid sensing" need modification.

The authors show endogenous cGAS is localized to the nucleus of HeLa cells by use of a monoclonal antibody in confocal micrographs. Specificity was verified by use of CRISPR'ed HeLa cells. This was extended to mouse cells using a different mAb, with specificity control being cGAS KO cells. Localization was verified with western blot analysis of nuclear and cytosolic extracts. Nuclear localization was independent of cGAS activation or cell cycle. Additional data support strong association that is salt dependent and the N-terminus is dispensable.

The data for nuclear localization are clear with excellent specificity controls. While it cannot be ruled out that there is a small amount of cGAS in the cytosol that mediates activation, the work certainly indicates that current models of cytosolic nucleic acid sensing are either incomplete and missing a step before cGAS becomes localized to the cytosol, or perhaps more likely, that it mediates sensing while in the cytosol itself. In either case, the paper raises important issues, even though it does not clearly address these latter points.

Reviewer #2:

Cyclic GMP-AMP synthase (cGAS), an innate immune sensor of double-stranded DNA was initially described as a cytosolic protein. However, increasing number of studies in the last 4 years indicate that this protein is also present in the nucleus. One of the key question in the field is how constant activation of nuclear cGAS by genomic DNA is voided to prevent immunopathology. In the present manuscript, Volkman et al. demonstrate that cGAS is present in the nucleus and is tightly tethered to chromatin. Based on this hypothesize that inactivation of nuclear cGAS is due to this tight tethering to chromatin. Further, they claim that this tethering is independent of the functional domains involved in cGAS activation. Their data showing the presence of cGAS in the nucleus looks convincing and expected given what has already been reported across several publication. The other key messages of the manuscript are however inconclusive and highly speculative. Overall, the observations reported in this manuscript are very preliminary. Most of their claims are at this time point of their analysis still speculative not supported by data. Many of the key experiments are also not properly designed and lack controls. Here below are some suggestions for improvement. I hope they find them useful.

Specific points:

1) Conceptual advancement over previous studies: The main message of this study i.e. the presence of cGAS in the nucleus has already been established by several previous studies (e.g. Orzalli et al., 2015;, Yang et al., 2017; Lahaye et al., 2018).

2) The proposal that the inactivity of nuclear cGAS is due to the tight tethering chromatin is interesting. However, this idea was not experimentally tested in this study. Besides, this idea is really not new and has been tested more rigorously in studies already in the public domain (Zierhut C and Funabiki H, bioRxiv, 2017, doi: 10.1101/168070).

3) The authors claim that tethering of cGAS to chromatin is independent of cGAS features important for its activation is not convincing. The functional features of cGAS required for its activation are mainly its DNA binding, enzymatic and dimerization activities. The functional behaviour of the cGAS truncations mutants cannot be assumed to be similar to that within the functional full-length protein. The structural-functional features of cGAS are well defined and a more convincing approach would be for the authors use cGAS point mutants defective in these features. It is really difficult to make any firm conclusion from the truncation mutants since essential controls are lacking. For example, what is the affinity of the different truncation mutants to DNA?

4) The K395M/K399M mutant of mouse cGAS used in Figure 4F is an inactive mutant. Is this because of defect in DNA binding mutant or dimerization? If author want to use enzyme inactive mutant of cGAS, the well characterized 225A/D227A mutant of human cGAS which is more appropriate.

5) The authors speculate that DNA is not the primary tether of cGAS but instead propose some protein component(s) of higher order chromatin. What is the experimental basis for this this proposal? In fact, their preliminary findings with the nuclease digestions suggest implicate DNA.

6) The authors claim that nuclear localization of cGAS is via active translocation rather than via passive mechanisms. While interesting, this proposal is also preliminary and require rigorous experimental controls. At the very minimums they should test whether blocking inhibitors of protein nuclear import prevent nuclear localization of cGAS.

7) Many of the experiments lack essential controls. For example, without loading controls for nuclear and cytosolic markers in the immunoblots in Figure 3, it difficult to interrogate the veracity of their claims.

[Editors’ note: what now follows is the decision letter after the authors submitted for further consideration.]

Thank you for submitting your article "Endogenous cGAS is predominantly a nuclear protein" for consideration by *eLife*. Your article has been reviewed by three peer reviewers, and the evaluation has been overseen by a Reviewing Editor and Tadatsugu Taniguchi as the Senior Editor. The following individuals involved in review of your submission have agreed to reveal their identity: Andrea Ablasser (Reviewer #3).

The reviewers have discussed the reviews with one another and the Reviewing Editor has drafted this decision to help you prepare a revised submission. Although nuclear localization of cGAS has already been established by published literature and that the current work leaves unclear the mechanistic basis and implications of cGAS’s nuclear localization, the consensus was that this manuscript is still relevant for the current discussion about the localization of cGAS. The reviewers have identified the following key points that need to be addressed before it can be considered further.

Summary:

cGAS, the main innate immune sensor of DNA inside cells is essential for immune defense against numerous pathogens, but has also been implicated in many inflammatory diseases. cGAS was initially assumed to be sequestered in the cytosol away from the genomic DNA in the nuclear compartments. However, an increasing number of studies indicate that cGAS is not only present in the cytosol but also in the nucleus and plasma membrane, raising the question, how nuclear cGAS is prevented from eliciting an immune reaction to self-DNA. Many of the previous investigations have mainly looked at overexpressed cGAS. In this study Volkman et al. have combined subcellular fractionation and microscopy to study the localization of endogenous cGAS. They show that endogenous cGAS is mainly in the nucleus where it is tightly tethered to chromatin. They propose that the inactivity of nuclear cGAS to genomic DNA is due to its tethering to chromatin. The points in this article are of general interested to researchers in the field of innate immunity.

Essential revisions:

1) A limitation of the paper is that the fractionation experiments are potentially susceptible to an artifact, which is that cGAS might associate with some dense structure that co-sediments with the nucleus, but is not actually itself nuclear. The potential for the fractionation experiments to be misleading means that the complementary immunofluorescence experiments (shown in Figure 1) are of critical importance. One main worry about the data in Figure 1 is that because the nucleus is thicker than the rest of the cell, it may appear that cGAS is concentrated in the nucleus simply because the confocal slices sectioned primarily through the nucleus, and missed the cytosol at the periphery of the cell; or, if a z-series projection of all slices is shown, the fact that there would be more slices through the nucleus might improve the signal for this part of the cell as compared to the cytosol. It is hard to tell exactly what was done because the Materials and methods and legends don't fully describe the imaging and analysis. This part of the paper should therefore be strengthened with some additional methodological description and controls. For example, was a Z-series collected? What step size? Are projections shown? Can a 3D reconstruction (x-z projection) of the cell be shown to confirm that the cGAS is truly nuclear? What is the volume of the nucleus versus the volume of the cytosol and is the apparent localization in the nucleus merely because the nucleus contains most of the cell volume? Can some kind of quantitation be performed on the images in Figure 1 (cytosol vs. nucleus)? Can a control cytosolic protein be shown to convince the reader that were cGAS cytosolic it would be detected? Is it surprising that the DAPI and cGAS images appear so similar? Why doesn't the so-called micronucleus in Figure 1B stain with DAPI (maybe it is not a micronucleus)?

2) One of the major conclusions of this study is that nuclear cGAS localization is not via passive association with chromatin (as reported by others e.g. Yang et al., 2017, Gentili et al., 2019) but via a more specific mechanism. The subcellular fractionation and mutant experiments are not sufficient to justify this conclusion. For example, in Figure 3F, in the absence of mitotic nuclear membrane dissolution, how can the presence of control GFP in their nuclear soluble (NS) and nuclear pellet (NP) fraction be explained? Due to passive or active import? Or could it be post-lysis contamination of such fractions with cytosolic components? Of course, GFP-cGAS is more abundant that control GFP in the NP fraction. However, this could simply be due to the ability of GFP-cGAS and not the GFP control to bind to DNA hence its retention therein. Therefore, similar to the above point, can the authors induce GFP-cGAS expression in cells arrested at G0 or G1 and visualize its localization by life confocal imaging to see distinctly that freshly synthesized cGAS does indeed translocate into the nucleus independent of mitotic nuclear membrane dissolution? In view of these caveats, the authors’ own admission that the nuclear import inhibitors tested did not impede nuclear localization (surprisingly not mentioned this in the text), the claim that nuclear cGAS is via active translocation remains very weak and is potentially misleading. Moreover, given the sensitivity of nuclear pellet-associated cGAS to nucleases (Figure 5) and since none of the cGAS mutants tested completely abrogates DNA binding, it is in simply not possible to discount the role genomic DNA in nuclear localization. Therefore, in the absence of direct evidence the authors should mellow down on this claim.

[Editors' note: further revisions were requested prior to acceptance, as described below.]

Thank you for submitting your revised work "Endogenous cGAS is predominantly a nuclear protein". The Reviewing Editor has now carefully gone through the manuscript and consulted with the Senior Editor. We are basically in favour of accepting the paper for *eLife*. Indeed, we appreciate your inclusion of the requested images to show more clearly the subcellular localization of cGAS. We also recognize the technical challenge of ruling out completely the role of DNA binding in cGAS nuclear localization and are willing to accept your explanation highlighted in the Discussion. However, for reasons described below, for us to consider this manuscript further the remaining experiment to shed more light into how cGAS localizes into the nucleus is still required.

As you are aware, cGAS nuclear localization has been reported across many publications already. Moreover, avid binding to chromatin as a possible mechanism for preventing cGAS-mediated immune response to nuclear DNA – one of the main points of the current manuscript, has also been described (Zierhut et al., Cell, 2019, formerly a preprint on bioRxiv). Therefore, in view of the current state of the field, the potentially new aspect of this manuscript is how cGAS is localized to the nucleus. Your theory that cGAS nuclear localization is via active transport from the cytosol advances an interesting concept with major implications in the field since this could provide new avenues for manipulating cGAS-mediated biological processes. On the other hand, if incorrect, this idea can mislead the field. As you have admitted in response to our earlier request, your nuclear import inhibitors experiments so far do not support your theory (these are relevant data and you may consider to include them in the manuscript). Although you have toned down on your initial conclusion, it remains unclear whether nuclear localization is indeed via active or non-specific mechanisms. Therefore at a minimum, the suggested live confocal imaging comparing the localization of inducible GFP-cGAS is still essential. This experiment is feasible and requires standard techniques that should be available in your institution or can be arranged through collaboration.

We previously brought to your attention the fact that GFP control alone does localize to the nucleus (Figure 3D). We have noted your explanation that such localization is likely because of passive translocation of GFP due to its size (27 kD). In this regard, it is equally important to note that passive diffusion across the nuclear envelope does not have fixed molecular mass threshold; proteins of molecular mass of up to 200 kD can diffuse across the nuclear envelope with varying kinetics (Timney et al., J. Cell Biol., 2016). Therefore, an alternative explanation perhaps worth considering is that nuclear cGAS localization is simply via passive diffusion. That is why live confocal imaging comparing the localization of inducible GFP-cGAS and a GFP-tagged control protein with similar molecular mass as cGAS would help to shed some light onto whether cGAS nuclear localization is indeed due to active or passive mechanisms. Even if the conclusion from this experiment were that localization is via passive mechanism, this would still be an important contribution and would provide more clarity to the field.

[Editors’ note: this article was then rejected after discussions between the reviewers, but the article was accepted after an appeal against the decision.]

Thank you for choosing to send your work entitled “Tight nuclear tethering of cGAS is essential for preventing autoreactivity" for consideration at *eLife*.

Your article and your letter of appeal have been considered by a Senior Editor, and we regret to inform you that we are upholding our original decision. Detailed comments by the Reviewing Editor are described below. I feel very sorry that we cannot be more positive at this stage and sincerely hope that the comments are valuable to your study.

This work was originally submitted under the title "Endogenous cGAS is predominantly a nuclear protein". The initial conclusion of the study were that: (1) cGAS is predominantly a nuclear protein, (2) that nuclear localization of cGAS is independent of DNA binding or cell cycle as previously reported by others, and (3) that this localization is due to active importation from the cytosol. The outstanding concerns throughout the previous rounds of revisions were that their data were inconsistent with the conclusion regarding mechanisms of nuclear localization and association with the chromatin.

In the previous decision letter, we requested one control experiment which in my opinion was feasible and hence the quickest path to publication. Specifically, to verify their claim that nuclear localization was via active mechanisms, we requested for life-microscopic images of their inducible cGAS. The authors did not however provide this controls experiment but have left out the data that we previously pointed out to be inconsistent with their conclusion (e.g. previously in Figure 3D). They have also included new data (not asked for) and have rewritten the manuscript in such a manner that the main message is now very different from the original submission.

In this version, the authors propose that at resting state, cGAS is kept inactive through tight binding (which the authors call tethering) to chromatin and that this tethering is via yet to be identified tether and not DNA. Implicit in this model is that for cGAS activation to occur, cGAS has to undergo untethering to enable it to bind DNA. They claim to have identified a conserved surface on cGAS responsible for this tethering and conclude that this surface is distinct from that for DNA binding. In my assessment the authors have misinterpreted their data and these fresh claims are misleading. In fact the new data support the opposite of their conclusion: that attachment of cGAS to the chromatin in via DNA.

Specific comments:

1) What the authors conclude as tethering surface lie within the DNA binding surface of cGAS. The R222E, R240E, R241E, R244E mutants that the authors report as tethering defective mutants are in fact DNA binding mutants. This is well established in the field (e.g. Figure 5C and Figure 6 of Li et al., Immunity, 2013). Moreover, the other mutants, for example the K335E, K382A, E386A, K395M, K399M which the authors used to conclude that tethering of cGAS to chromatin is independent of DNA binding have also DNA binding mutants. The main difference these sets of mutants is that the suggested "tethering mutants" (R222E, R240E, R241E, R244E) are severely defective in DNA binding compared to the K335E, K382A, E386A, K395M, K399M mutants which retain substantial DNA binding (Figure 6 of Li et al., Immunity, 2013). Therefore in my view, the correct interpretation is that interaction of cGAS with chromatin involves some form of interaction with DNA binding. This is consistent with the authors' data showing that cGAS-chromatin interaction is highly sensitive to DNases (Figure 5).

2) Related to the above point, in my view, the more plausible explanation for the spontaneous activity of the R222E, R240E, R241E, R244E mutants is that mutations in these amino acids likely triggers a conformational change lowering the threshold of activation by DNA. And, of course the compound mutations in R222E together with either K335E, K382A or K395M is expected to result in an inactive mutant (Figure 6A, Figure 4F), since the latter mutations are in themselves inactivating mutation (Li et al., 2013).

3) The manuscript contains statements that are potentially misleading to the readers. For example in the Introduction the authors state "Here, we use confocal microscopy and biochemical characterization to determine the resting localization of endogenous cGAS prior to activation". In the Discussion, the authors go on to conclude, "We show, using microscopy and biochemical fractionation, that the great majority of endogenous cGAS is nuclear prior to its activation." Inherent in this statement is that in resting state cGAS is kept inactive through interaction with chromatin but then undergoes spatial redistribution upon activation. There is no evidence that this is the case.

4) The authors proposed tethering models and the Discussion largely assume that all/or most of the cGAS activation occurs in the nucleus, how do the authors explain cGAS activation following DNA transfection or following some bacterial infections – that such sensing also occurs in the nucleus or that this involves redistribution of cGAS to the cytosol? A shortcoming of the proposed model is that it does factor the constant presence of a chromatin-free cGAS and that it is this pool that most likely becomes activated by foreign or misplaced self-DNA?

In brief, this revised manuscript has not addressed previous concerns. Whereas their data showing that cGAS is abundantly nuclear is convincing and consistent with those reported by recent studies, as exemplified above, many of the key conclusions, especially the new ones, are in my assessment not supported by the data. Therefore I unfortunately I cannot recommend the manuscript for publication. My suggestion to the authors would be to include the suggested controls and refocus the manuscript to the original message and instead describe/characterize the new autoactive mutants more coherently in separate manuscript.

---

## [Author Response]

[Editors’ note: the author responses to the first round of peer review follow.]

Reviewer #1:This manuscript shows that endogenous cGAS, a nucleic acid sensor, is predominately localized to the nucleus. This observation suggests that current models of "cytosolic nucleic acid sensing" need modification.The authors show endogenous cGAS is localized to the nucleus of HeLa cells by use of a monoclonal antibody in confocal micrographs. Specificity was verified by use of CRISPR'ed HeLa cells. This was extended to mouse cells using a different mAb, with specificity control being cGAS KO cells. Localization was verified with western blot analysis of nuclear and cytosolic extracts. Nuclear localization was independent of cGAS activation or cell cycle. Additional data support strong association that is salt dependent and the N-terminus is dispensable.The data for nuclear localization are clear with excellent specificity controls. While it cannot be ruled out that there is a small amount of cGAS in the cytosol that mediates activation, the work certainly indicates that current models of cytosolic nucleic acid sensing are either incomplete and missing a step before cGAS becomes localized to the cytosol, or perhaps more likely, that it mediates sensing while in the cytosol itself. In either case, the paper raises important issues, even though it does not clearly address these latter points.

We thank the reviewer for the assessment of the manuscript, and we agree that the resting localization of endogenous cGAS needs to be taken into account when considering its mechanism of activation. Our study provides clear evidence that the cytosolic DNA sensing model is insufficient to explain the transition between resting and activated cGAS.

Reviewer #2:[…]Specific points:1) Conceptual advancement over previous studies: The main message of this study i.e. the presence of cGAS in the nucleus has already been established by several previous studies (e.g. Orzalli et al., 2015;, Yang et al., 2017; Lahaye et al., 2018).

We thank the reviewer for highlighting these papers that we have already cited in our submitted manuscript, together with the Funabiki bioRxiv preprint that we did not cite because it is not yet a peer-reviewed publication. None of these papers, nor any of the published studies on cGAS and micronuclei, suggest that cGAS is predominantly and constitutively a nuclear protein as we show here. To summarize their findings relative to what we show in our manuscript.

Orzalli MH et al:

* Importantly, in data that were in Figure 4H of the original manuscript and are now included as supplemental data, we show that IFI16 plays no role in cGAS tethering.

Yang et al:

*Our data clearly demonstrate that the great majority of endogenous cGAS is nuclear, in all cells tested, and independent of mitosis. If the model that cGAS redistributes to the cytosol during interphase were true, then arresting cells in interphase should cause cGAS relocalization. In Figure 3 of our manuscript, we show that cells arrested for three days in interphase do not relocalize endogenous cGAS to the cytosol.

Lahaye et al:

*In this study, a fraction of cGAS was found to be in the nucleus, although the authors only sampled the low salt nuclear extract and did not identify the salt-resistant pool of endogenous cGAS that comprises the majority of nuclear cGAS across nine different cell types. The authors also presented evidence that NONO is required for the nuclear localization of this low salt-extractable cGAS. We now present data in a new supplemental figure showing endogenous cGAS localization in four independent clonal lines of NONO-deficient HeLa cells. NONO deficiency does not influence the tight tethering of cGAS in the nucleus. We acknowledge in the manuscript that differences in cell types could influence the importance of NONO in controlling cGAS nuclear localization, but we emphasize that NONO is not required for nuclear tethering of cGAS.

Zierhut and Funabiki:

*This manuscript follows the Yang et al. proposal that cGAS is cytosolic, associates with mitotic chromatin, and then redistributes to the cytosol during the next interphase. We elaborate on the data in this manuscript in point 2 below.

2) The proposal that the inactivity of nuclear cGAS is due to the tight tethering chromatin is interesting. However, this idea was not experimentally tested in this study. Besides, this idea is really not new and has been tested more rigorously in studies already in the public domain (Zierhut and Funabiki, 2017, bioRxiv, doi: 10.1101/168070).

In the Funabiki manuscript, the authors incubate recombinant cGAS with purified nucleosomes. They show, interestingly, that recombinant cGAS binds to these purified nucleosomes slightly more tightly than it binds to naked DNA. However, none of these studies were done with endogenous cGAS in actual cells. Instead, they were done in a test tube, with salt and solute concentrations that do not equate to those inside live cells. We propose that our analyses of endogenous cGAS in eight different cell lines are relevant, they are rigorous, and they reveal new and interesting biology. Our future goals are to define the nature of the tether, perturb it, and examine the consequences for cGAS localization and activation. However, these experiments are outside the scope of our current manuscript that describes cGAS nuclear tethering for the first time.

3) The authors claim that tethering of cGAS to chromatin is independent of cGAS features important for its activation is not convincing. The functional features of cGAS required for its activation are mainly its DNA binding, enzymatic and dimerization activities. The functional behaviour of the cGAS truncations mutants cannot be assumed to be similar to that within the functional full-length protein. The structural-functional features of cGAS are well defined and a more convincing approach would be for the authors use cGAS point mutants defective in these features. It is really difficult to make any firm conclusion from the truncation mutants since essential controls are lacking. For example, what is the affinity of the different truncation mutants to DNA?4) The K395M/K399M mutant of mouse cGAS used in Figure 4F is an inactive mutant. Is this because of defect in DNA binding mutant or dimerization? If author want to use enzyme inactive mutant of cGAS, the well characterized 225A/D227A mutant of human cGAS which is more appropriate.

There are a number of important points here and we welcome the opportunity to address them with new data.

First, defined truncation mutants are a well-established approach to explore isolated regions and domains of proteins required for a specific function or interaction. We used this approach to define the region(s) of cGAS that are required to mediate its tight tethering in the nucleus. Notably, we clearly show that the unstructured N-terminus, which is essential for DNA-induced condensation and cGAS activation, is completely dispensable for cGAS nuclear localization and tethering.

Second, the K395M/K399M murine cGAS is the corresponding mutant to the K407A/411A mutant of human cGAS described in Civril et al., 2013. This mutant disrupts key lysines in the DNA-binding platform, its DNA binding is reduced but not absent, and it is completely defective for cGAMP production, as shown in Figure 4F of our revised manuscript and in their original description of the mutant.

Third, we now include five additional point mutants in Figure 4 of the revised manuscript to test DNA binding and DNA-induced dimerization/oligomerization. We cloned and validated these mutants, transudced them into human fibroblasts using the dox-inducible lentivirus system, induced with dox, measured resting and activated cGAMP production, and determined nuclear localization and tethering. These experiments with defined mutants provide compelling evidence that the mechanism of cGAS activation by DNA is distinct from the mechanism of cGAS tethering. We cannot claim that cGAS tethering is completely independent of DNA binding; such a claim would be impossible to test because proving that a particular mutant of cGAS is completely inert for DNA binding is not currently possible.

5) The authors speculate that DNA is not the primary tether of cGAS but instead propose some protein component(s) of higher order chromatin. What is the experimental basis for this this proposal? In fact, their preliminary findings with the nuclease digestions suggest implicate DNA.

In the Discussion section of the manuscript, we attempt to reconcile the known weak affinity of cGAS for DNA with the remarkably tight tethering of nuclear cGAS. The new mutants suggested by the reviewer and discussed in point 4 above make the question more relevant since we use defined, biochemically characterized mutants that are defective for DNA binding and DNA-induced oligomerization and show that cGAS remains tethered. We suggest that DNA binding alone cannot explain the tight tethering. Moreover, we offer the logical consideration that if nuclear cGAS were directly bound to DNA, two problems arise. First, how would DNA-bound cGAS remain inactive? Second, how would cGAS that is saturated with self DNA be competent to respond to foreign DNA? Such a model would require titration of cGAS off of self DNA and onto foreign DNA, at which point it would become active. We speculate that non-DNA components of higher order chromatin are essential for cGAS tethering. We are not attempting to prove that with the data in our paper; instead, we are proposing a model that can explain both the known properties of cGAS and our new findings.

6) The authors claim that nuclear localization of cGAS is via active translocation rather than via passive mechanisms. While interesting, this proposal is also preliminary and require rigorous experimental controls. At the very minimums they should test whether blocking inhibitors of protein nuclear import prevent nuclear localization of cGAS.

This is a good suggestion. We attempted to use two different inhibitors of nuclear import: ivermectin, which inhibits importin α/β-dependent nuclear import (Wagstaff et al., Biochem J, 2012), as well as importazole, which inhibits importin β-dependent nuclear import (Soderholm et al., ACS Chem Biol, 2011). These inhibitors only block a fraction of nuclear import mechanisms, and there are numerous pathways of nuclear import for which there are no adequate inhibitors. We found that neither of these drugs, alone or in combination, prevented cGAS nuclear localization in arrested cells, and that they were both extremely toxic to the cells. We modify the description of these data to state only that cGAS nuclear localization is more complicated than simple passive association with chromatin during mitosis, and we remove the statement suggesting active import. We emphasize that our cell cycle analyses are the first to rigorously test the prevailing model that cGAS is cytosolic during interphase.

7) Many of the experiments lack essential controls. For example, without loading controls for nuclear and cytosolic markers in the immunoblots in Figure 3, it difficult to interrogate the veracity of their claims.

We apologize for not including these loading controls in the original figure, and this is an important point because we switch from the sequential salt washes in Figure 2 to the three-fraction separation in Figure 3. We now include loading controls monitoring cytosol, nuclear supernatant, and nuclear pellets, the same proteins that are followed as controls for Figure 2; this allows more specific comparison of the two approaches.

[Editors' note: the author responses to the re-review follow.]

Essential revisions:1) A limitation of the paper is that the fractionation experiments are potentially susceptible to an artifact, which is that cGAS might associate with some dense structure that co-sediments with the nucleus, but is not actually itself nuclear. The potential for the fractionation experiments to be misleading means that the complementary immunofluorescence experiments (shown in Figure 1) are of critical importance.

We emphasize that among the recently published studies evaluating the cellular distribution of cGAS, our study is the only one in which the microscopy of endogenous cGAS agrees perfectly with the biochemical fractionations. We have presented microscopy data for endogenous cGAS in two cell types with two distinct fixation methods. We have performed biochemical analysis of endogenous cGAS in six different cell types from mouse and human, and we have employed seven defined cGAS mutants to test whether nuclear localization and tight tethering require robust DNA binding, DNA-activated dimerization, ‘phase separation,’ and catalytic activity.

One main worry about the data in Figure 1 is that because the nucleus is thicker than the rest of the cell, it may appear that cGAS is concentrated in the nucleus simply because the confocal slices sectioned primarily through the nucleus, and missed the cytosol at the periphery of the cell; or, if a z-series projection of all slices is shown, the fact that there would be more slices through the nucleus might improve the signal for this part of the cell as compared to the cytosol. It is hard to tell exactly what was done because the Materials and methods and legends don't fully describe the imaging and analysis. This part of the paper should therefore be strengthened with some additional methodological description and controls. For example, was a Z-series collected? What step size? Are projections shown? Can a 3D reconstruction (x-z projection) of the cell be shown to confirm that the cGAS is truly nuclear?

We have updated the Materials and methods to describe in more detail the confocal microscopy and the z-series step size. Additionally, we have added two videos to Figure 1: a combined rendering of all the z-stacks through an image, and a 3D rendering of the merged z-stacks. Together, these data clearly demonstrate that the great majority of endogenous cGAS is in the nucleus.

What is the volume of the nucleus versus the volume of the cytosol and is the apparent localization in the nucleus merely because the nucleus contains most of the cell volume? Can some kind of quantitation be performed on the images in Figure 1 (cytosol vs. nucleus)?

Volumetric measurements of the entire nucleus and cytosol are outside the scope of this manuscript and require more sophisticated tools and analysis that we do not currently use. Importantly, the biochemical analysis of cGAS and direct quantitation of cGAS abundance by densitometry in Figure 2B offer a quantitative comparison of the amount of cGAS in the cytosol versus the nucleus. Again, we emphasize that the microscopy of endogenous cGAS agrees perfectly with the biochemistry.

Can a control cytosolic protein be shown to convince the reader that were cGAS cytosolic it would be detected?

We have now included a three-color image of endogenous cGAS, tubulin, and DAPI in Figure 1A, together with z-series and 3D renderings as supplemental movies. This demonstrates that an abundant cytosolic protein is preserved and detected with our fixation and staining methods.

Is it surprising that the DAPI and cGAS images appear so similar? Why doesn't the so-called micronucleus in Figure 1B stain with DAPI (maybe it is not a micronucleus)?

We realized that the images presented in Figure 1 were dimmer than ideal in the rendered pdf, so we re-analyzed the images to uniformly brighten the cGAS and DAPI fluorescence. The micronucleus-like structure in Figure 1B is indeed DAPI positive, and the brightened images now show this clearly.

2) One of the major conclusion of this study is that nuclear cGAS localization is not via passive association with chromatin (as reported by others e.g. Yang et al., 2017, Gentili et al., 2019) but via a more specific mechanism. The subcellular fractionation and mutant experiments are not sufficient to justify this conclusion. For example, in Figure 3F, in the absence of mitotic nuclear membrane dissolution, how can the presence of control GFP in their nuclear soluble (NS) and nuclear pellet (NP) fraction be explained? Due to passive or active import? Or could it be post-lysis contamination of such fractions with cytosolic components? Of course, GFP-cGAS is more abundant that control GFP in the NP fraction. However, this could simply be due to the ability of GFP-cGAS and not the GFP control to bind to DNA hence its retention therein.

GFP, because of its size (27 kD), is very well known to be able to enter the nucleus through passive diffusion through the nuclear pore, without any requirement for energy-dependent nuclear import (Timney et al., J. Cell Biol., 2016). The distribution of GFP alone compared GFP-cGAS in Figure 3D is dramatically different. While there is a very small amount of GFP remaining in the nuclear pellet, the localization of the great majority of GFP-cGAS to the pellet and its absence from the low salt nuclear supernatant are the most important points here. Again, the fractionation of GFP-cGAS agrees with the fractionation of endogenous cGAS.

Therefore, similar to the above point, can the authors induce GFP-cGAS expression in cells arrested at G0 or G1 and visualize its localization by life confocal imaging to see distinctly that freshly synthesized cGAS does indeed translocate into the nucleus independent of mitotic nuclear membrane dissolution?

Live confocal imaging of GFP-cGAS requires tools and analyses that we do not currently have, and we do not think that they would add substantively to the data presented throughout the manuscript. In Figure 3C-D, we arrest cells at the G1/S border, induce cGAS with dox, and show that it localizes to the nucleus identically to the localization in cycling cells. Importantly, this experiment definitively demonstrates that mitosis is not required for cGAS nuclear localization. The next question of precisely how cGAS enters the nucleus is one that will require more work, but it is not relevant for the conclusion that we draw based on the data in the manuscript. We have changed the last sentence of the paragraph describing Figure 3 to focus on this important point and to not speculate further.

Original sentence: “These data demonstrate that cGAS nuclear localization is not mediated by its passive association with chromatin during mitosis, and they suggest a more specific mechanism by which cGAS enters the nucleus.”

Revised sentence: “These data demonstrate that mitosis is not required for cGAS nuclear localization.”

In view of these caveats, the authors’ own admission that the nuclear import inhibitors tested did not impede nuclear localization (surprisingly not mentioned this in the text), the claim that nuclear cGAS is via active translocation remains very weak and is potentially misleading.

The nuclear import inhibitors that are commercially available only block a fraction of specific modes of nuclear import, whereas some forms of nuclear import cannot be chemically inhibited.

Moreover, given the sensitivity of nuclear pellet-associated cGAS to nucleases (Figure 5) and since none of the cGAS mutants tested completely abrogates DNA binding, it is in simply not possible to discount the role genomic DNA in nuclear localization. Therefore, in the absence of direct evidence the authors should mellow down on this claim.

We tested the contributions of robust DNA binding, DNA-activated dimerization, and DNA-activated ‘’phase-separation” using structure-guided mutants that all dramatically reduce the DNA-mediated activation of full-length cGAS in live cells. All of these mutants remain nuclear and remain tethered. We agree that none of the cGAS mutants tested (nor any that have ever been made) completely abrogate DNA binding. Importantly, the ability of cGAS or its mutants to bind DNA has always been measured with truncated, recombinant cGAS lacking its N terminus, under salt and solute conditions that do not adequately resemble the cellular environment, and with short double-stranded synthetic DNAs that do not fully recapitulate the size and disposition of DNA within cells. We feel that our data, with defined mutants, support the conclusions that we draw and our interpretation of the findings.

In the Discussion section of the manuscript, we attempt to rationalize our findings using seven defined cGAS mutants, together with the fact that chromatin is required for cGAS tethering. We interpret our data and offer a proposal based on our novel findings, with a very important logical consideration that has not been discussed in the literature. We feel that this is an extremely important point, that we have not over-interpreted our data, that it is appropriate for inclusion in the Discussion section, and that it is worth presenting because it offers a new model for cGAS regulation. To emphasize that it is impossible to completely rule out DNA binding, we have re-worded this section of the Discussion:

“While we cannot completely rule out a role for DNA binding in cGAS nuclear tethering because no known cGAS mutant is completely devoid of DNA binding, we speculate that DNA itself is not the primary tether of cGAS. […] Based on these considerations, we speculate that nuclear cGAS does not directly interact with genomic DNA, and that the tethering mechanism maintains resting cGAS in a state that is competent to detect foreign DNA.”

[Editors' note: further revisions were requested prior to acceptance, as described below.]

As you are aware, cGAS nuclear localization has been reported across many publications already. Moreover, avid binding to chromatin as a possible mechanism for preventing cGAS-mediated immune response to nuclear DNA – one of the main points of the current manuscript, has also been described (Zierhut et al., Cell, 2019, formerly a preprint on bioRxiv). Therefore, in view of the current state of the field, the potentially new aspect of this manuscript is how cGAS is localized to the nucleus. Your theory that cGAS nuclear localization is via active transport from the cytosol advances an interesting concept with major implications in the field since this could provide new avenues for manipulating cGAS-mediated biological processes. On the other hand, if incorrect, this idea can mislead the field. As you have admitted in response to our earlier request, your nuclear import inhibitors experiments so far do not support your theory (these are relevant data and you may consider to include them in the manuscript). Although you have toned down on your initial conclusion, it remains unclear whether nuclear localization is indeed via active or non-specific mechanisms. Therefore, at a minimum, the suggested live confocal imaging comparing the localization of inducible GFP-cGAS is still essential. This experiment is feasible and requires standard techniques that should be available in your institution or can be arranged through collaboration.We previously brought to your attention the fact that GFP control alone does localize to the nucleus (Figure 3D). We have noted your explanation that such localization is likely because of passive translocation of GFP due to its size (27 kD). In this regard, it is equally important to note that passive diffusion across the nuclear envelope does not have fixed molecular mass threshold; proteins of molecular mass of up to 200 kD can diffuse across the nuclear envelope with varying kinetics (Timney et al., J. Cell Biol., 2016). Therefore, an alternative explanation perhaps worth considering is that nuclear cGAS localization is simply via passive diffusion. That is why live confocal imaging comparing the localization of inducible GFP-cGAS and a GFP-tagged control protein with similar molecular mass as cGAS would help to shed some light onto whether cGAS nuclear localization is indeed due to active or passive mechanisms. Even if the conclusion from this experiment were that localization is via passive mechanism, this would still be an important contribution and would provide more clarity to the field.

We thank the reviewers and the editors for their comments. At issue throughout this revision and resubmission process is a potential “mechanism” for whether, how, and when cGAS enters the nucleus. These experiments are asked for in the context of questions over whether our findings are real or an elaborate artifact, because cytosolic DNA sensing is the foundational framework of the field. It has been suggested that our fixation conditions for microscopy might result in the loss of a relevant pool of cytosolic cGAS and an overestimate of the amount of nuclear cGAS (Barnett et al., 2019). Moreover, it has also been suggested that our extraction conditions might cause cytosolic cGAS to artifactually condense on DNA liberated during extraction, which would make it co-sediment with the nuclei during centrifugation that separates cytosol from nucleus (Barnett et al., 2019). We had addressed both of these concerns with experiments in our prior revised manuscript, but they still remain. Finally, although we were the first to describe tight nuclear tethering of cGAS, its relevance remained unknown and was only speculated on in the Discussion section of our last submission.

Since the original submission of our paper to *eLife* in December 2018, four new reports have been published that address cGAS localization and its relevance, each arriving at distinct conclusions.

1) Barnett et al., Cell 2019 (PMID 30827685) suggest that cGAS is predominantly a cytosolic protein anchored to the plasma membrane via interactions between its N-terminus and PIP2.

2) Zierhut et al., Cell 2019 (PMID 31299200) suggest that cGAS is cytosolic but associates with mitotic chromatin after nuclear envelope breakdown. They suggest that purified nucleosomes mildly inhibit activation of truncated cGAS (lacking its N-terminus) in a test tube but they do not demonstrate anything about tethering in cells.

3) Gentili et al., Cell Reports 2019 (PMID 30917330) suggest that cGAS localizes to specific regions of the genome, and that its N terminus is important for nuclear localization.

4) Jiang et al., EMBO 2019 (PMID 31544964) suggest that cGAS is bound to chromatin through its DNA-binding residues, and that cGAS condensation onto genomic DNA inhibits DNA repair.

In new data presented in a new Figure 6, we have identified the surface of cGAS that is responsible for its tight nuclear tethering. This surface, and key amino acids within it, are conserved in every known vertebrate cGAS protein spanning 350 million years of evolution. We mutated single conserved amino acids of cGAS within this surface. This results in untethering and elution of cGAS in low salt, but the mutant cGAS proteins remain predominantly nuclear. Remarkably, these single amino acid mutations cause massive, constitutive, DNA-dependent activation of cGAS. Some of these mutant cGAS enzymes are so active upon expression alone that they cannot produce any more cGAMP in response to DNA transfection. Thus, untethered cGAS assembles on and is saturated by self-DNA, resulting in maximal activation. Finally, the location of this tethering surface, which is in the NTase domain (not the N-terminus), is such that binding of cGAS to a tethering protein would be incompatible with DNA binding. Thus, tethering and DNA-based activation of cGAS are mutually exclusive states.

By identifying the tethering surface of cGAS, disrupting it, and observing the consequences of untethering, we:

– prove that cGAS is indeed predominantly a nuclear protein that is tightly tethered to chromatin (this has been questioned as an artifact of fixation or lysis conditions).

– corroborate all of our mutant data in Figure 4 showing that DNA binding, oligomerization, condensation, and catalysis are all dispensable for nuclear localization and tethering.

– demonstrate that other recent studies suggesting that the N-terminus of cGAS is required for cGAS localization to either the plasma membrane (Kagan) or nucleus (Manel) are incorrect.

– reveal that tethering physically sequesters cGAS from genomic DNA in the nucleus, distinct from recently proposed models that invoke DNA binding as the mechanism of nuclear localization (e.g. Gekara, Funabiki, Manel).

– show that tethering is essential for preventing cGAS assembly on self-DNA.

– reveal that in order for cGAS to be activated, tethering must be overcome. Thus, cGAS is carefully regulated *prior* to its encounter with DNA, and there must be a regulated step to permit cGAS activation by foreign DNA.

– identify a fundamental and evolutionarily conserved mechanism that underlies self/non-self discrimination by cGAS.

We feel that these new and exciting findings not only corroborate the existing data in our paper, but they also reveal a fundamental new mechanism for preventing nuclear cGAS from responding to nuclear genomic DNA. As such, we think that these data also obviate any lingering concerns on the part of the reviewers and editors over whether nuclear tethering is real and relevant.

We include several additional pieces of data in order to address any residual concerns over our microscopy images:

1) New microscopy images in which we compare methanol fixation to paraformaldehyde fixation for visualization of endogenous nuclear cGAS.

2) Imaging flow cytometry analysis in which we quantitate nuclear versus cytosolic GFP-cGAS in thousands of live, unfixed cells.

[Editors’ note: the author responses to the final round of editors’ comments follow.]

This work was originally submitted under the title "Endogenous cGAS is predominantly a nuclear protein". The initial conclusion of the study were that: (1) cGAS is predominantly a nuclear protein, (2) that nuclear localization of cGAS is independent of DNA binding or cell cycle as previously reported by others, and (3) that this localization is due to active importation from the cytosol.

We are unaware of any papers, published prior to our submission on December 3, 2018, that have demonstrated that cGAS nuclear localization is independent of DNA binding or cell cycle, but would welcome any references we may have overlooked.

The outstanding concerns throughout the previous rounds of revisions were that their data were inconsistent with the conclusion regarding mechanisms of nuclear localization and association with the chromatin.In the previous decision letter, we requested one essential control experiment which in my opinion was feasible and hence the quickest part to publication. Specifically, to verify their claim that nuclear localization was via active mechanisms, we requested for life-microscopic images of their inducible cGAS. The authors did not however provide this control experiment but have left out the data that we previously pointed out to be inconsistent with their conclusion (e.g. previously in Figure 3D). They have also included new data and have rewritten the manuscript in such a manner that the main message is now very different from the original submission.

Two important points here. First, the main point of our manuscript is, and always has been, that cGAS is a tightly tethered nuclear protein. When we submitted these findings to *eLife* on December 3, 2018, this was absolutely a novel finding. In our subsequent revisions, we included fifteen additional cGAS mutants to support this, all of which are completely congruent with our main point.

Second, we agreed with the prior reviewer's assessment that our claim that cGAS import was an active process was not fully supported by our data. We removed this from the revised manuscript because the experimentation required to adequately test it would not be incisive or quantitative. Importantly, chemical inhibitors that block all forms of active nuclear import do not exist.

Indeed, the question of *how* cGAS gets into the nucleus is tangential to the important and novel observation that cGAS is in fact predominantly nuclear, and that it is tightly tethered in the nucleus. For this reason, we decided to include the mechanistic data that unequivocally define the tethering surface of cGAS, together with the first description of the consequences of untethering. These data absolutely support the original message of the manuscript and do not change the original point that cGAS is a tightly tethered nuclear protein. In fact, the new data demonstrate the actual importance of tight nuclear tethering. We believe that proving the "mechanism" of cGAS nuclear import is beyond the scope of this manuscript, and we instead defined the cGAS nuclear tethering surface and its essential role in preventing cGAS autoreactivity. In addition, we believe our new data in Figure 6 supports, enriches, and extends the main point of our original manuscript.

In this version, the authors propose that at resting state, cGAS is kept inactive through tight binding (which the authors call tethering) to chromatin and that this tethering is via yet to be identified tether and not DNA. Implicit in this model is that for cGAS activation to occur, cGAS has to undergo untethering to enable it to bind DNA. They claim to have identified a conserved surface on cGAS responsible for this tethering and conclude that this surface is distinct from that for DNA binding. In my assessment the authors have misinterpreted their data and these fresh claims are misleading. In fact, the new data support the opposite of their conclusion: that attachment of cGAS to the chromatin in via DNA.Specific comments:1) What the authors conclude as tethering surface lie within the DNA binding surface of cGAS. The R222E, R240E, R241E, R244E mutants that the authors report as tethering defective mutants are in fact DNA binding mutants. This is well established in the field (e.g. Figure 5C and Figure 6 of Li et al., Immunity, 2013). Moreover, the other mutants, for example the K335E, K382A, E386A, K395M, K399M which the authors used to conclude that tethering of cGAS to chromatin is independent of DNA binding have also DNA binding mutants. The main difference these sets of mutants is that the suggested "tethering mutants" (R222E, R240E, R241E, R244E) are severely defective in DNA binding compared to the K335E, K382A, E386A, K395M, K399M mutants which retain substantial DNA binding (Figure 6 of Li et al., Immunity, 2013). Therefore, in my view, the correct interpretation is that interaction of cGAS with chromatin involves some form of interaction with DNA binding. This is consistent with the authors' data showing that cGAS-chromatin interaction is highly sensitive to DNases (Figure 5).

There are some factual errors here:

R222E was defined as a severe DNA binding mutant in the Li et al. paper, but it was also shown to be fully active upon transient transfection in which the plasmid DNA also serves as a potent activating ligand. We explain this clearly and cite the Li paper extensively.

K240E (not R240E) was also defined as a DNA binding mutant in the Li paper but was also paradoxically fully active upon transient overexpression.

R241 is actually not a DNA binding residue. It points away from DNA in the crystal structure, and it was in fact never tested in the Li paper. Thus, R241 is not part of the DNA-binding surface of cGAS.

R2444 is not a "DNA-binding" residue. It is actually unresolved in the crystal structure, and it was also not tested in the Li paper. Perhaps the editor is accidentally referring to the R342E mutant, which is 98 amino acids away from R244.

As such, to say that all the mutants that we "report as tethering defective mutants are in fact DNA binding mutants" is incorrect, and we hope the editor will reconsider this point upon revisiting Li et al.

2) Related to the above point, in my view, the more plausible explanation for the spontaneous activity of the R222E, R240E, R241E, R244E mutants is that mutations in these amino acids likely triggers a conformational change lowering the threshold of activation by DNA. And, of course the compound mutations in R222E together with either K335E, K382A or K395M is expected to result in an inactive mutant (Figure 6A, Figure 4F), since the latter mutations are in themselves inactivating mutation (Li et al., 2013).

R222E and K240E were defined as DNA binding mutants in a test tube, with recombinant cGAS lacking its N terminus, under salt conditions that do not resemble those in cells. If these are severe DNA binding mutants, why are they massively and constitutively activated in a DNA-dependent manner? Instead, R222E, K240E, R241E and the additional mutants that we define are constitutively active. The well-defined DNA binding residues that compromise cGAS function when mutated show that the constitutive activation of R222E is DNA-dependent. Importantly, as shown in Figure 4, K335E, K395M/399M, zinc thumbless, K382A, E386A are all still tethered in the nucleus. This is amply supported by the seventeen distinct mutants we describe, all of which have been added in our revised submissions to put our model to a rigorous test.

3) The manuscript contains statements that are potentially misleading to the readers. For example, in the Introduction the authors state "Here, we use confocal microscopy and biochemical characterization to determine the resting localization of endogenous cGAS prior to activation". In the Discussion the authors go on to conclude, "We show, using microscopy and biochemical fractionation, that the great majority of endogenous cGAS is nuclear prior to its activation." Inherent in this statement is that in resting state cGAS is kept inactive through interaction with chromatin but then undergoes spatial redistribution upon activation. There is no evidence that this is the case.

With five mouse cGAS mutants and two human cGAS mutants, we show that there is a dramatic redistribution of constitutively active cGAS from high salt fractions to low salt fractions. The DNA-dependent activation of these constitutive mutants is accompanied by a redistribution into lower salt fractions. If DNA is primarily responsible for tightly tethering inactive, resting cGAS, it seems unlikely that the highly constitutive, DNA-dependent, autoreactive cGAS would elute at much lower salt concentrations.

4) The authors proposed tethering models and the Discussion largely assume that all/or most of the cGAS activation occurs in the nucleus, how do the authors explain cGAS activation following DNA transfection or following some bacterial infections – that such sensing also occurs in the nucleus or that this involves redistribution of cGAS to the cytosol? A shortcoming of the proposed model is that it does factor the constant presence of a chromatin-free cGAS and that it is this pool that most likely becomes activated by foreign or misplaced self-DNA?

As Figure 6 demonstrates that the untethered mutants of cGAS remain nuclear and are activated by self-DNA, we hope it is clear that the cytosol is not the only place that cGAS can be activated by DNA.